# The ecological and developmental foundations of brood parasitism in a catfish

Martin Reichard [1,2,3] ✉, Radim Blažek[1,3], Matej Polačik[1], Tomáš Suchánek [4], Gernot K. Englmaier[1], Kacper Pyrzanowski [1,2], Veronika Bartáková[1], Jakub Žák[3], Lukas Koch [1,3], Iva Dyková[3], Robert Cerny [4], Stephan Koblmüller [5] & Holger Zimmermann [1,5] ✉

Interspecific brood parasitism has evolved repeatedly from parental care. However, many non-avian brood-parasitic lineages have ancestors lacking parental care, leaving the foundations of brood parasitism in these lineages enigmatic. We examine ecological, reproductive, and developmental data from the Lake Tanganyika radiation of *Synodontis* catfishes, where one species, the cuckoo catfish, exhibits brood parasitism of mouthbrooding cichlid fishes. Our comparative analyses, coupled with experimental parasitism, suggest that a combination of ancestral traits (large eggs and rapid embryo development) enabled the origin of brood parasitism. Evolutionary innovations then presumably enhanced the success of brood parasitism after it emerged. Novel traits comprise frequent production of small clutches to effectively utilize host availability, the evolution of egg mimicry to facilitate host egg adoption, and modifications to development to enhance the performance of catfish embryos in the host's buccal cavity. Interestingly, despite the distinct ecological and life history contexts of the origin of catfish brood parasitism, its evolutionary and developmental outcomes align with those of canonical avian brood parasites. This suggests that general patterns are repeated in the evolution of brood parasitism, despite disparate starting conditions.

A loudly begging cuckoo chick, tirelessly fed by a pair of tiny songbird hosts, serves as a compelling illustration of brood parasitism. Obligate brood parasites—species that usurp the parental care of other species—have independently evolved seven times in birds, with over 100 bird species classified as obligate brood parasites[1]. Theoretical and empirical insights from research on brood parasitism have influenced our understanding of coevolutionary dynamics[1–6] and life history evolution[7,8], though this research has been largely confined to avian model systems where parental care is ancestral to brood parasitism[6]. However, interspecific brood parasitism is more ubiquitous in nature[1,6], with one-third (19 of 59) of cases found in lineages lacking

parental care, with a transition to brood parasitism instead preceded by trophic ecological relationships such as saprophagy, scavenging and parasitoidy[9]. Hence, to better understand the conditions that facilitate the evolution and persistence of brood parasitism, it is imperative to examine the origins and early stages of brood parasitism in lineages that lack parental care.

In this study, we combined ecological, reproductive, and developmental datasets across the Lake Tanganyika (LT) *Synodontis* catfish (Siluriformes, Mochokidae) radiation along with additional riverine *Synodontis* species to examine the origin and early evolution of brood parasitism in the cuckoo catfish, *Synodontis multipunctatus*. While

[1]Institute of Vertebrate Biology, Czech Academy of Sciences, Brno, Czech Republic. [2]Department of Ecology and Vertebrate Zoology, University of Lodz, Lodz, Poland. [3]Department of Botany and Zoology, Faculty of Science, Masaryk University, Brno, Czech Republic. [4]Department of Zoology, Faculty of Science, Charles University, Prague, Czech Republic. [5]Institute of Biology, University of Graz, Graz, Austria. ✉e-mail: reichard@ivb.cz; holger.zimmermann@uni-graz.at

other *Synodontis* species scatter their eggs over the substrate without parental care[10], the cuckoo catfish is an obligate brood parasite of mouthbrooding LT cichlid fishes[11]. Many LT cichlids tend their eggs and embryos in their buccal cavity (termed mouthbrooding)[12]. Cuckoo catfish exploit this advanced mode of parental care by intruding on spawning cichlid pairs and spawn themselves[13,14]. As adult cuckoo catfish are also vigorous egg predators[15], the spawning female cichlid quickly responds to the risk of egg predation by hastily picking up her eggs, inadvertently collecting some eggs deposited by the intruding cuckoo catfish[16]. Cuckoo catfish offspring is then brooded in safety alongside the cichlid eggs in the female's mouth. The catfish eggs develop faster than those of the host and, after hatching, feed on the host's offspring[17,18]. This system resembles the relationship between cuckoos and their hosts, with the host deceived into caring for the cuckoo's offspring and incurring a significant cost to its own reproductive success. Given that adult cuckoo catfish are potent egg predators, we hypothesized that the origin of brood parasitism in the cuckoo catfish has primarily arisen from the predation of adult catfish on cichlid eggs during spawning[19]. Here, we investigate the ecological, reproductive, and life history background of the LT *Synodontis* to identify traits associated with the origin and evolution of brood parasitism in *Synodontis*.

The cuckoo catfish is part of a small radiation of LT *Synodontis*[20,21]. This lineage consists of two clades (Fig. 1a). The first (Clade I) comprises the cuckoo catfish (*S. multipunctatus*) and deepwater (depth >40 m) *Synodontis granulosus*. The second (Clade II) includes *Synodontis petricola*, *Synodontis irsacae*, and *Synodontis polli*, all commonly found in rocky habitats at depths from 5 m alongside the cuckoo catfish[22,23]. The age of the LT *Synodontis* radiation has been estimated to be 2.7 (1.7–3.9) million years ago (Mya), with similar dating for further diversification within Clade I (1–2 Mya) and Clade II (0.9–1.9 Mya)[24].

We demonstrate that cuckoo catfish are trophic generalists rather than specialized egg predators. Their reproductive traits combine ancestral and novel characteristics that facilitate brood parasitism and consist of small batches of large, non-adhesive eggs that appear to mimic the large yellow eggs of their hosts. Developmental adaptations of the cuckoo catfish include enlarged jaws, pharyngeal teeth and associated skeletal supports, as well as accelerated mineralization of their dentition and supporting skeleton, enabling them to grasp and puncture host offspring and feed on their nutrient-rich yolk inside the cichlid buccal cavity. Many of these traits likely evolved in other contexts but were later co-opted to enhance the brood parasitic strategy of the cuckoo catfish.

## Results

### Brood parasitism in the cuckoo catfish is not associated with trophic specialization

Brood parasitism may evolve from a trophic relationship between parasitic and host species, including the exploitation of host eggs[9]. Furthermore, many evolutionary radiations originated from dietary specialization[12,25,26]. Thus, we examined gastrointestinal tracts and employed stable isotopic analysis to investigate whether the cuckoo catfish occupies a specific trophic niche within LT *Synodontis*. Considering that host egg predation is a feasible stepping stone to interspecific brood parasitism[9,19], we hypothesized that the cuckoo catfish specializes to feed on nutritionally rich cichlid eggs.

First, we analyzed the diets of all five LT *Synodontis* species collected in the lake. Dissecting the gastrointestinal tracts of 122 specimens and calculating Bray–Curtis dissimilarities (BCDs) based on volumetric estimates of their diet revealed that the primary differences among LT *Synodontis* species were associated with the relative contribution of sponges, benthic insect larvae, fish tissue, and algae (Fig. 1b, Likelihood Ratio tests: Table S1, SIMPER analysis: Table S2). The diet of the cuckoo catfish quantitatively differed from those of all

other LT *Synodontis* species (multivariate GLM, $P < 0.001$; Table S2), and uniquely amongst LT *Synodontis*, the gut of one (of 30 examined) cuckoo catfish indeed contained fish eggs. However, the small size (<1 mm), spherical shape and high abundance ($n = 257$) of consumed eggs clearly indicated the eggs did not belong to any cichlid species confirmed to be host of the cuckoo catfish[27]. The cuckoo catfish is obviously not a specialized egg predator. Instead, its diet is very generalized; its gastrointestinal tracts contained a mix of dietary components, featuring a high proportion of benthic aquatic invertebrates from soft sediments (Fig. 1b). This is despite that three other LT *Synodontis* species are clear dietary specialists (Fig. 1b); the variation in their within-species BCD values is negligible compared to between-species BCDs (<0.32, Fig. 1c and Table S4). The diet of *S. granulosus* consisted solely of fish tissue (bones and ctenoid scales of either cichlids or *Lates*, Fig. S1)[28,29], typical of predators and scavengers. In contrast, *S. polli* is an herbivore, feeding on algae and plant detritus. The third specialist (*S. irsacae*) fed primarily on sponges (Porifera) (Fig. 1b, c). Like the cuckoo catfish, *S. petricola* is identified as a dietary generalist (Fig. 1b, c and Table S4).

The examination of gastrointestinal tracts offers only a short time window for food consumption. To compare time-integrated trophic ecology, we analyzed signatures from stable isotopic (SI) ratios of $\delta^{13}C$ and $\delta^{15}N$ in muscle tissues from 198 catfish. The results confirmed substantial trophic niche separation among LT *Synodontis* (Fig. 1d, and Table S5). All species differed in their trophic position, except for the similar SI signatures observed in the cuckoo catfish and *S. petricola* (PERMANOVA: Table S6), two trophic generalists. The trophic level of *S. granulosus* was higher than that of a specialist predatory cichlid (Fig. 1e), suggesting that *S. granulosus* is a fish scavenger rather than a predator[30]. The herbivorous *S. polli* encompassed the combined niches of two herbivorous cichlids[12] and sponge-eating *S. irsacae* occupied an extreme dietary niche with no known LT cichlid equivalent (Fig. 1e). The SI trophic niche of the cuckoo catfish was broadest of all LT *Synodontis* species (Fig. 1d). When compared to the trophic niche space of the cichlid radiation in Lake Tanganyika, which is strongly underpinned by trophic ecology[12], the cuckoo catfish largely overlapped with the SI space of several cichlid species (Fig. 1e). This further discounts the hypothesis of its trophic specialization on fish eggs, which would be associated with a higher $\delta^{15}N$ signature[31].

Both diet and SI analyzes consistently demonstrate a trophic component in the LT *Synodontis* radiation. However, the generalized diet of the cuckoo catfish and the rare occurrence of fish eggs in its diet undermine our hypothesis that the cuckoo catfish is a specialized egg predator. While the detection window for eggs in the gastrointestinal tract is short[32] and may lead to underestimation, the gastrointestinal tracts of West African *Synodontis* species frequently contain fish eggs[33]. Therefore, strict trophic specialization is not a key driver in the origin of brood parasitism in this lineage.

### The reproductive traits of cuckoo catfish combine ancestral and novel characteristics that facilitate brood parasitism

One of the biggest challenges for brood parasites is to synchronize their reproduction with that of their hosts and to induce them to accept parasitic eggs. We predicted that the cuckoo catfish produces multiple small clutches to match the continuous breeding season of their hosts[34] and thus be ready to capitalize on forthcoming opportunities for parasitism. We also predicted that cuckoo catfish have evolved egg characteristics that facilitate host acceptance through mimicry in terms of egg size and coloration. The usual *Synodontis* spawning is seasonal and involves egg scattering, during which the eggs may adhere to a substrate[10]. Since egg adherence may hinder collection by host cichlids and might compromise developmental performance in the host's buccal cavity, we predicted decreased egg adhesiveness of cuckoo catfish eggs. We used hormonally-induced reproduction in five LT *Synodontis* and three additional riverine

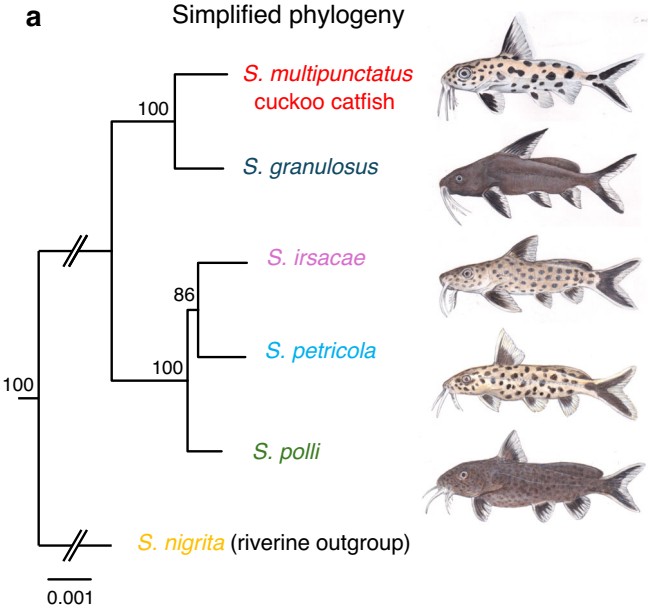

**a** Simplified phylogeny

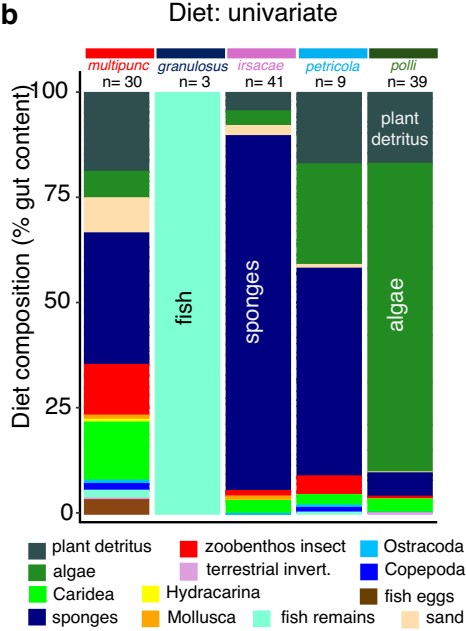

**b** Diet: univariate

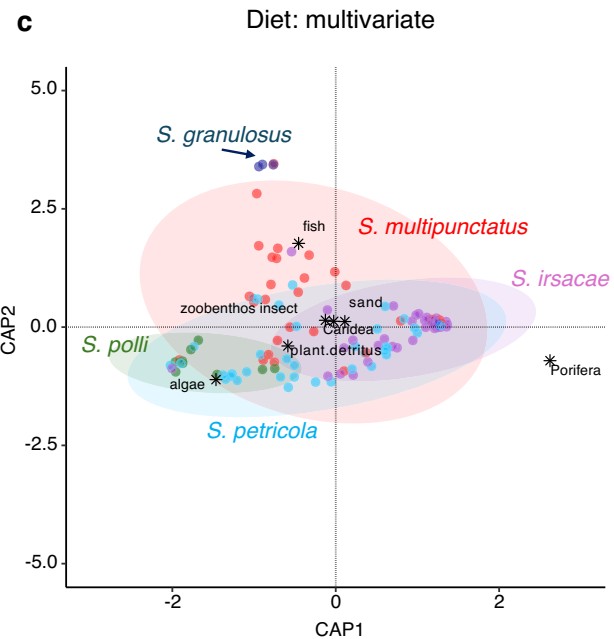

**c** Diet: multivariate

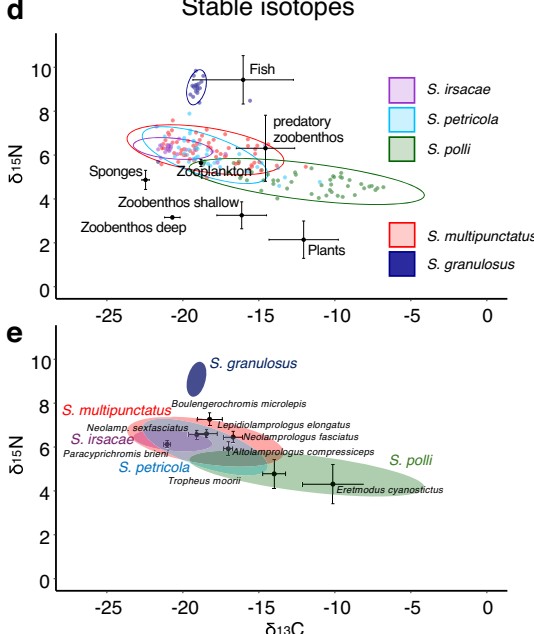

**d** Stable isotopes

**e**

**Fig. 1 | Phylogenetic relationships, dietary, and trophic separation among LT *Synodontis*. a** The maximum likelihood phylogenetic tree of the studied LT *Synodontis* flock, based on the concatenated alignment of 6536 ddRADseq loci and 5000 ultrafast bootstrap replicates[25]. **b** The diet composition of LT *Synodontis* expressed as the volumetric proportion of specific dietary categories in the gastrointestinal tract (n = 122; species-specific sample size is provided above each bar). **c** A biplot from the principal coordinates analysis (PCoA from the 'vegan' R package) comparing diet composition across LT *Synodontis*, derived from Bray–Curtis dissimilarity matrices. Ellipses represent the 95% confidence intervals for each species. Black dots denote the positions of key diet components that differentiate the species-specific diet. **d** Iso-space plots of LT *Synodontis* (n = 198) plotted over mean (95% CI) values of baseline items (n = 34). Values are reported in standard permil notation (‰); long-term analytical precision was 0.2‰ for δ13C values and 0.1‰ for δ15N values. **e** Overlap between coexisting LT *Synodontis* (95% CI ellipses) and selected cichlid species (mean and 95% CI error bars) (n = 62). Source data are provided as a Source Data file.

*Synodontis* species to collect data on standardized clutch and egg characteristics. We further obtained histological data on the gonads of four LT *Synodontis* species from individuals collected in the wild at the start and end of the rainy season (n = 239).

Notably, the clutch size of the cuckoo catfish (mean 28, range 12-45 eggs) was an order of magnitude smaller than that of all other *Synodontis* species (means of 208-767 eggs) (LM, P < 0.001, Fig. 2a and Table S7). The size of individual cuckoo catfish eggs (2.7 mm) was approximately double that of other LT (1.3–1.8 mm) and riverine (1.3–1.5 mm) *Synodontis* (LM, P < 0.001), except for its sister species, *S. granulosus*, which produced slightly larger eggs (2.8 mm) (Fig. 2b and Table S8). *Synodontis* egg size was not related to body size, as the cuckoo catfish (female total length 83 ± 1.9 mm) and *S. petricola* (87 ± 1.9 mm) were smaller than *S. irsacae* (99 ± 3.0 mm), *S. polli* (120 ± 1.7 mm) (Fig. 2c, d) and riverine *Synodontis* [10], as well as *S. granulosus* (131 mm), the largest LT *Synodontis* species. The color of

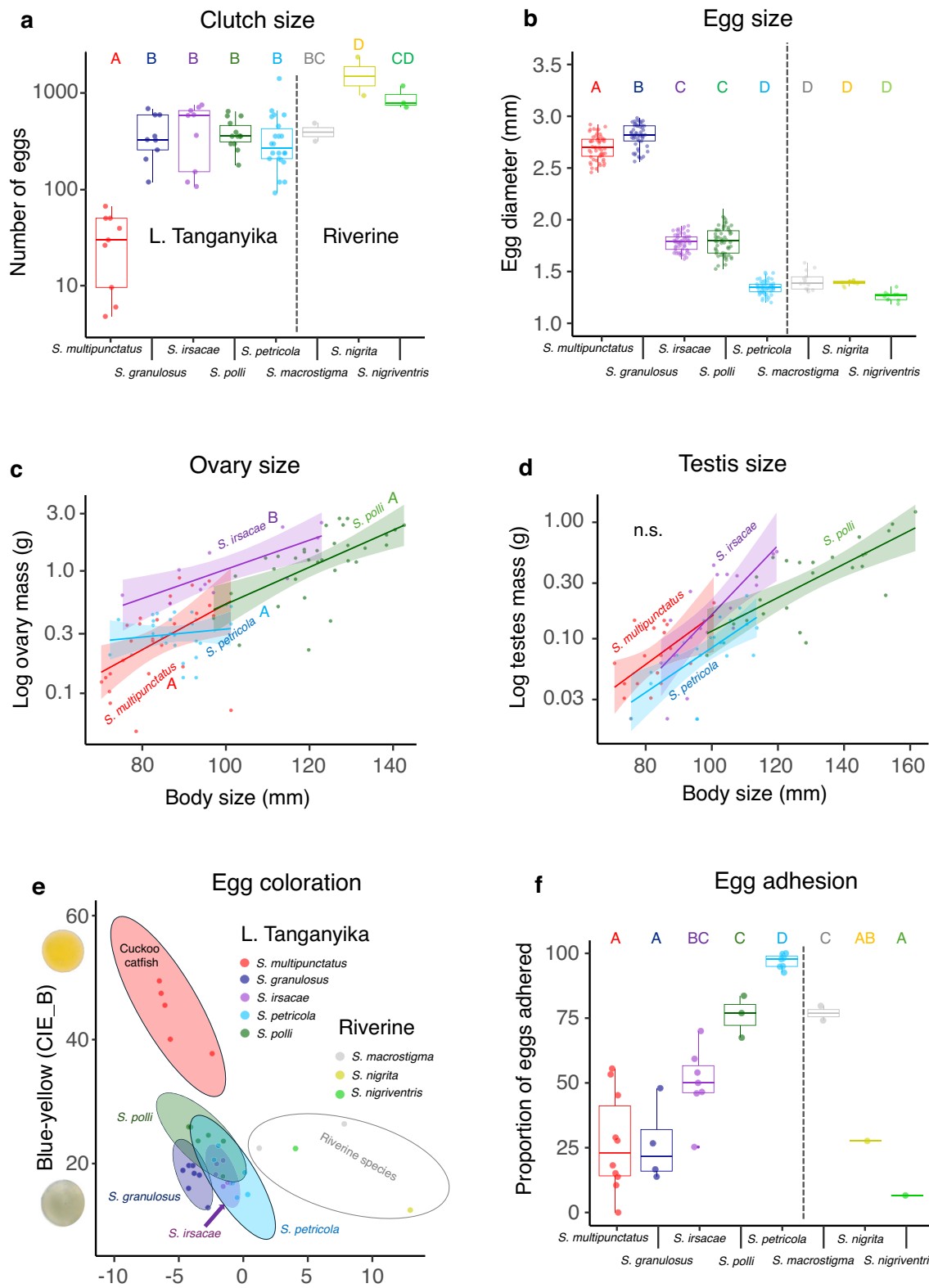

cuckoo catfish eggs was distinctly shifted towards a yellow hue compared to the more transparent eggs of all other LT *Synodontis* and the opaque brownish eggs of riverine *Synodontis* (LM, yellow-blue CIE_B axis: *P* < 0.001; red-green CIE_A axis: *P* ≤ 0.03; Fig. 2e and Table S9). This modification of the cuckoo catfish egg color and size appears to imitate notably larger, yellow eggs of host cichlids[34–36]. Finally, while the eggs of three LT *Synodontis* were highly adhesive (50–95%

remained adhered to a vertical surface), the cuckoo catfish and *S. granulosus* laid eggs with considerably lower adhesiveness (20-25% adhered; ordered-beta GLM, *P* < 0.001). Low adhesiveness was not exclusive to the cuckoo catfish and *S. granulosus*, as it was also recorded in some riverine *Synodontis* (Fig. 2f and Table S10). We also conducted ancestral trait reconstructions and calculated evolutionary signal metrics (Pagel's λ and Blomberg's K) for reproductive traits.

**Fig. 2 | Reproductive parameters of LT and riverine *Synodontis*. a** Clutch size, measured as the number of ripe eggs extracted from female gonads following artificial reproduction via hormonal stimulation ($n = 80$, two-sided LM, $\chi^2 = 297$, df = 7, $P < 0.001$). **b** Egg size (measured along the longest axis of IVF-produced freshly spawned eggs) ($n = 340$ eggs from 34 clutches, two-sided LMM, $\chi^2 = 2010$, df = 7, $P < 0.001$). **c** Ovary mass (grams, log-transformed) ($n = 127$, GLM, $\chi^2 = 21.3$, df = 3, $P < 0.001$) and **d** Testes mass, expressed as a product of fish body size (mm) in individuals with mature gonads ($n = 118$, GLM, $\chi^2 = 4.8$, df = 3, $P = 0.185$). Raw data points, best fit linear models with 95% confidence intervals are displayed. **e** Egg coloration, measured using cone-catch images transformed into CIELab color space, with the first two axes representing green-red (CIE-A) and blue-yellow (CIE-B). Insets on the left side display photographs of the eggs of *S. granulosus* and *S. multipunctatus*. **f** Egg adhesion, expressed as a percentage of freshly laid eggs remaining adhered to a vertical glass surface ($n = 35$ clutches, ordered-beta GLM, $\chi^2 = 119$, df = 7, $P < 0.001$). The box plots show median (line), 25th and 75th percentiles (box), and $1.5 \times$ the interquartile range (whiskers). Different letters in **a**–**c**, and **f** indicate statistically different values (Post-hoc tests with Benjamini–Hochberg correction, $P < 0.05$; Tables S7–S10). No further adjustments for multiple comparisons were made. All replicates were biological, not technical, replicates. Source data are provided as a Source Data file.

Significant phylogenetic signal was detected in egg coloration along the red–green axis, but not the yellow–blue axis, and moderate phylogenetic signal was observed for clutch size (Fig. S2).

Reproductive seasonality, based on the histological examination of the gonads of fish collected in the lake, was observed in a single species, *S. irsacae*, with 0% of females and 40% of males exhibiting ripe gonads in November and 93% and 100% in March ($n = 56$; $\chi^2$ tests: $P < 0.001$ and $P = 0.003$). In contrast, other LT *Synodontis* had 67–90% of fish with ripe gonads across sexes and species in November, and 94-100% in March ($n = 182$ across three species; Table S11). The relative female investment in reproduction, calculated as ovary mass relative to body size in fish with ripe gonads, was highest in the seasonally spawning *S. irsacae* (GLM, $P < 0.05$), but comparable among other LT *Synodontis*, including the cuckoo catfish (Fig. 2c). Relative testis mass did not differ among species (GLM, $P = 0.185$; Fig. 2d).

The analysis of reproductive traits thus revealed that cuckoo catfish produce small batches of large, non-adhesive eggs that mimic the large yellow eggs of their hosts[34] which likely facilitates egg adoption by the hosts. Further, the large cuckoo catfish eggs contain large yolk (nutrients) reserves to support rapid early development[17]. The regular production of small egg batches uniquely enables the cuckoo catfish to spawn every few days[14], while other traits conducive to brood parasitism (large egg size, low egg adhesiveness, and lack of reproductive seasonality) are not specific adaptations to a parasitic strategy but rather preceded its origin.

## The early development of the cuckoo catfish has evolved to exploit the host clutch

In the host buccal cavity, cuckoo catfish hatch early and, upon consumption of their own yolk sac, prey on the host clutch[17]. We predicted that cuckoo catfish embryos possess adaptations for predation on host offspring inside the cichlid buccal cavity, specifically in terms of the accelerated development of their dentition and jaw apparatus[17]. We hormonally induced the reproduction of all five LT and two riverine *Synodontis* species and produced a standardized fine-scale ontogenetic series following in vitro fertilization. We then compared early growth and development, with particular emphasis on the oral and pharyngeal dentition.

All *Synodontis* species hatched within the first 2 days post fertilization (dpf) and consumed their embryonic yolk sac at 6 dpf (Fig. S3). The cuckoo catfish and *S. granulosus* exhibited higher post-hatching growth rates than all other species (Fig. 4a). The offspring of these two species were nearly twice as large (LM, $P < 0.001$, Table S12) and considerably more developmentally advanced at the age of 10 dpf (Fig. 4b). However, only the cuckoo catfish possessed a well-mineralized dentition at 6 dpf (Fig. 4c) when they started to consume host offspring[17,18]. The cuckoo catfish also initially possessed larger jaws (Fig. 4c), pharyngeal teeth, and associated skeletal support (Figs. 4d and S4), as well as accelerated mineralization of its dentition and supportive skeleton (Table S13) in comparison to other LT *Synodontis*, including *S. granulosus*. *Synodontis granulosus* initially had a slightly larger pharyngeal dentition than other species (Fig. S4, and

Table S13), but the pace of its development and mineralization did not differ from the other species.

Comparison with the rate of dental development in other teleost fishes clarified that the development of the cuckoo catfish's dentition and supportive skeleton is strongly accelerated[37,38]. We propose that the cuckoo catfish has evolved a uniquely rapid post-hatching development, which enables its embryos to grasp and puncture host offspring to feed on their nutrient-rich yolk. These adaptive modifications of the cuckoo catfish derive from pre-existing traits, such as larger eggs and a more rapid post-hatching growth, which it shares with its sister species *S. granulosus*. However, *S. granulosus* do not possess dental and skeletal adaptations to feed on host embryos. These observations corroborate our hypothesis that the embryo development of the cuckoo catfish is under strong selection to enhance its ability to consume host clutches, resulting in the rapid development of embryonic dentition.

## Potential for brood parasitism in *S. granulosus*

Considering that large egg size and rapid embryo growth are shared between the cuckoo catfish and its sister species *S. granulosus*, it might be speculated that *S. granulosus* represents an unrecognized brood parasite. Unlike all other LT *Synodontis* species, which are commonly bred in captivity and - with the exception of the cuckoo catfish - documented as egg scatterers[10,39], there is a lack of data on the reproduction of deepwater *S. granulosus*. Hence, we examined whether *S. granulosus* eggs and embryos can survive, develop, and feed in the buccal cavity of a host cichlid and repeated the same experiment with *S. polli* as a non-parasitic species control.

We experimentally inoculated cichlids[13] with *S. granulosus* eggs and tested their survival over the incubation period of 4, 7, 12 and 14 days, using the cuckoo catfish eggs inoculated into other mouth-brooding females as controls. First, we used a Lake Victoria cichlid (*Haplochromis* sp. CH44), a non-sympatric cichlid species which has not evolved egg rejection of the cuckoo catfish[13]. We found that the survival rate of *S. granulosus* in mouthbrooding *Haplochromis* was high (75% of 56 eggs inoculated across 15 mouthbrooding host females and all incubation periods), and consistent with survival of the cuckoo catfish embryos (73.2% of 56 eggs across 14 mouthbrooding hosts; Binomial GLMM, $P = 0.873$) (Fig. 4a, b). The ability to survive incubation in cichlid mouths is not a universal feature of LT *Synodontis*; no embryos of non-parasitic *S. polli* were recovered from *Haplochromis* hosts (20 eggs across 5 mouthbrooding hosts) after 7 days of incubation, whereas 93% (26 of 28 inoculated eggs across 7 mouthbrooding hosts) cuckoo catfish control embryos survived that period (Fig. 4c). Cuckoo catfish embryos grasp and pierce host embryos to suck their yolk sacs (Fig. 4d, e) from day 4 post hatching and no cichlid embryos were found in experimentally parasitized females ($n = 13$) at age of 12–14 days. Despite having a comparable body size at that age (Fig. 3a, b), the dentition of *S. granulosus* embryos is relatively undeveloped (Fig. 3c, d), and the alimentary tract of *S. granulosus* offspring recovered from host buccal cavity contained only traces of material resembling yolk (Fig. 4f) while host offspring also disappeared ($n = 9$ host clutches).

**a**  Growth curves of *Synodontis* offspring

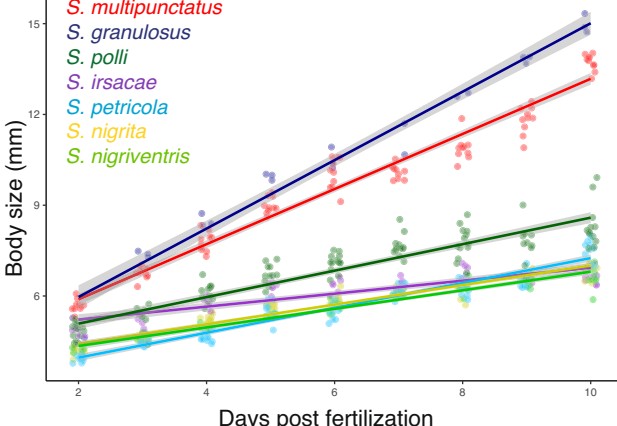

**b**  Appearance of *Synodontis* species 10 dpf

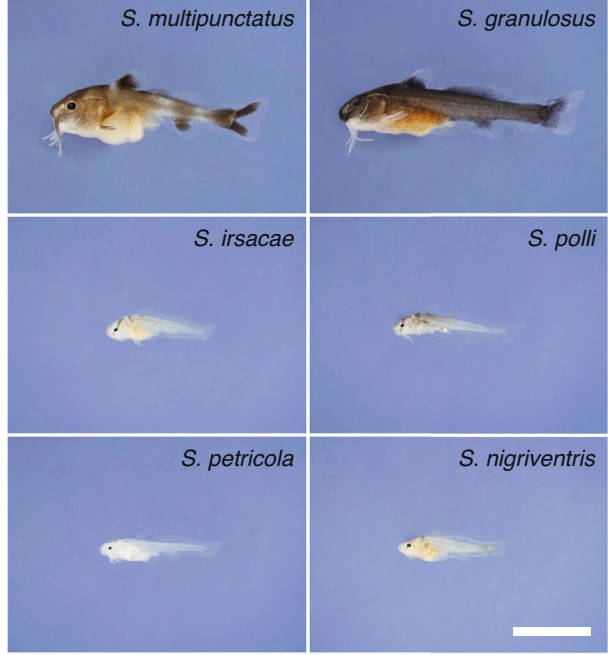

**c**  Oral dentition

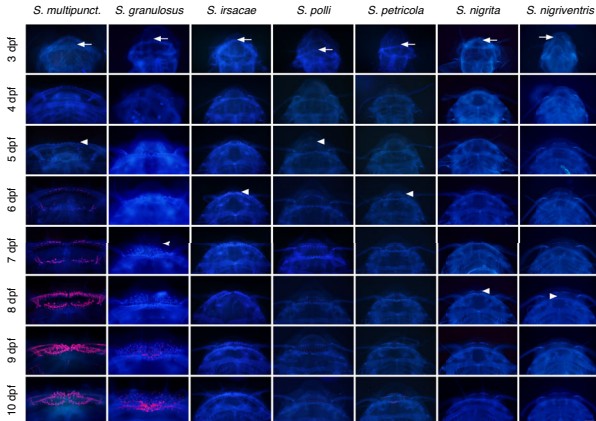

**d**  Pharyngeal dentition

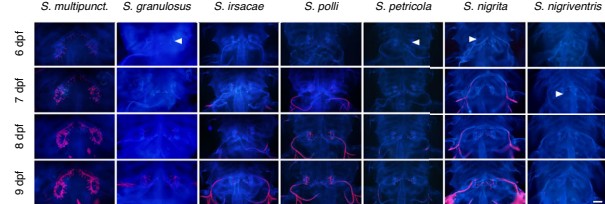

**Fig. 3 | Early development of *Synodontis* embryos. a** Growth (notochord length) of seven *Synodontis* species (visualized as a linear model fitted for post-hatching growth with 95% confidence intervals and raw data points, 2–10 days post-fertilization, n = 558). Source data are provided as a Source Data file. **b** The appearance of all LT (and one outgroup) *Synodontis* species at the age of 10 days, presented to the same scale. First appearance (arrow) and mineralization (red color) of (**c**) oral and (**d**) pharyngeal dentition in seven *Synodontis* species (scale bar in each inset photo). The complete series of pharyngeal dentition ontogeny is shown in Fig. S4.

When *S. granulosus* eggs were inoculated into the buccal cavity of a natural cuckoo catfish host (*Shuja horei*)[27] (which can effectively decrease the load of cuckoo catfish eggs[40]), none of the 18 eggs (across 6 mouthbrooding *S. horei*) survived 4 or 7 days of incubation, compared to a 30% survival rate of cuckoo catfish eggs (6 of 20 eggs across 4 of 6 mouthbrooding *S. horei*) (Fig. 4g). This finding suggests that the apparent exaptations (i.e., adaptations originally evolved outside parasitism context) to brood parasitism in *S. granulosus* are sufficient to survive 14 days of incubation (i.e., 8 days after yolk sac consumption) in an evolutionarily naive host, although their capacity to extract resources from the host clutch was inefficient. However, these putative exaptations to brood parasitism failed to support survival in the more challenging environment of an LT cichlid which is a common host of the cuckoo catfish[27] and has evolved defenses against parasitism[17].

If *S. granulosus* is indeed an unrecognized brood parasitic species parasitizing deep-water cichlids, it possesses many traits of the generalized *Synodontis* life history. First, its very large clutches exceed the capacity of the host buccal cavity by more than an order of magnitude. Second, the eggs do not visually imitate host eggs. Third, their ovulation frequency is not adapted to the opportunistic exploitation of unpredictable opportunities to parasitism. Finally, its embryonic dentition is not adapted to efficient extraction of resources from host embryo, as seen in the cuckoo catfish. Nevertheless, by using hosts from shallow-water habitats in our experiment, we cannot rule out the possibility that *S. granulosus* has specialized to parasitise deep-water cichlids. Irrespectively of the breeding strategy of *S. granulosus*, its reproductive and developmental characteristics appear to represent ancestral conditions within this small clade and perhaps exemplify the transient state, with many traits which have evolved in a different context, and from which brood parasitism in cuckoo catfish might have evolved.

## Discussion

Brood parasitism has evolved multiple times in species whose ancestors exhibited no parental care. It has been hypothesized that trophic exploitation of the potential host brood (e.g., predation, parasitoids) initiates the origin of brood parasitism[9], but it remains unclear whether the subsequent host-parasite coevolutionary arms race converges on traits analogous to those of canonical brood parasites[8]. With the limitation of testing a single evolutionary origin without a replication[41], we investigated this hypothesis in the cuckoo catfish, a brood parasite of mouthbrooding cichlids. Although

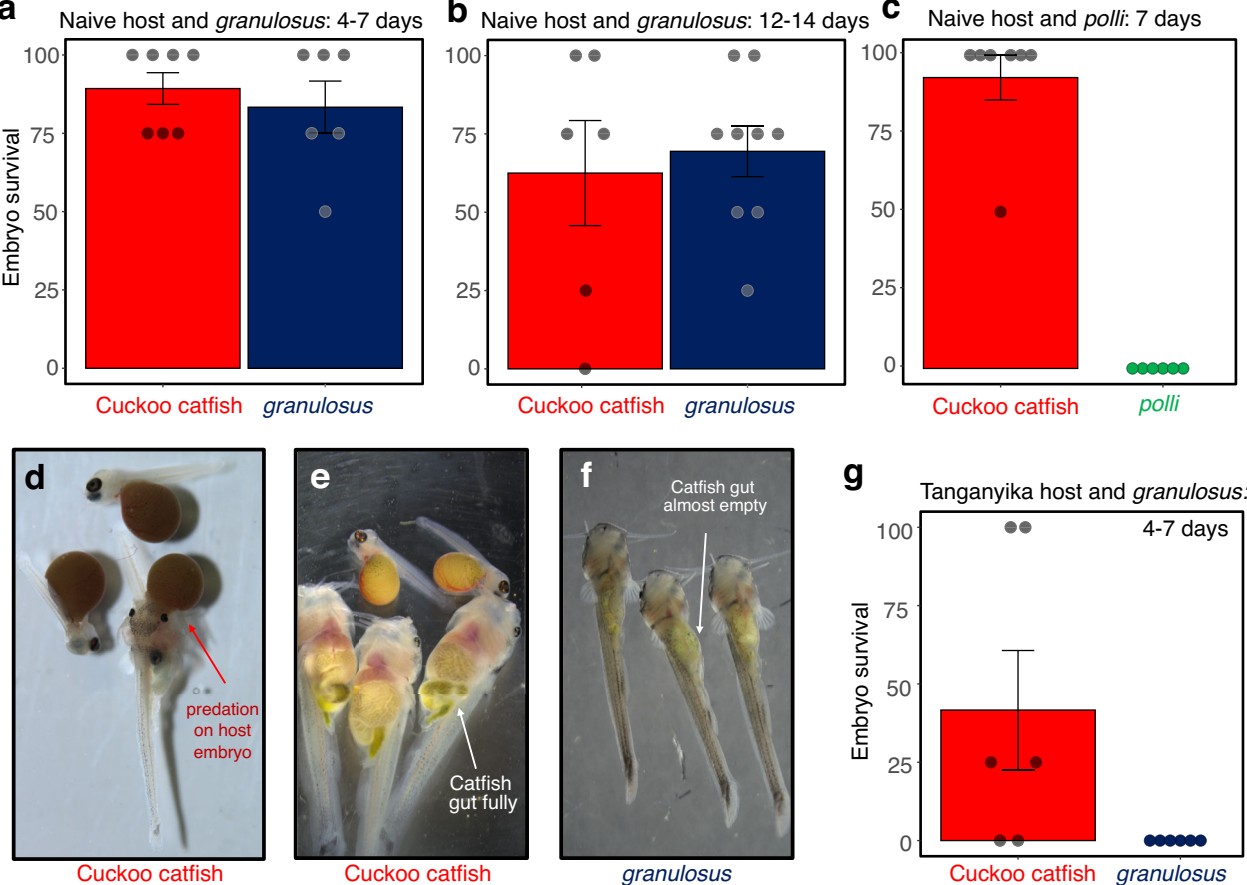

**Fig. 4 | Potential for brood parasitism in *S. granulosus* and *S. polli*.** Proportion of surviving embryos of *S. granulosus*, *S. polli* and their respective cuckoo catfish controls in an evolutionary naive non-sympatric (non-rejecting) host cichlid (*Haplochromis* CH44 from Lake Victoria with an egg diameter of 2.8 mm) over (**a**), the period of 4–7 days ($n = 20$ eggs across 6 clutches for *S. granulosus* and 28 eggs across 7 clutches in the cuckoo catfish controls), **b** 12–14 days ($n = 36$ eggs across 9 clutches for *S. granulosus* and 24 eggs across 6 clutches in the cuckoo catfish controls) and (**c**) the period of 7 days for *S. polli* treatment ($n = 40$ eggs across 10 clutches for *S. polli* and 56 eggs across 14 clutches in the cuckoo catfish controls). **d** A cuckoo catfish embryo consuming host offspring, with the host's eye visible in

the catfish's stomach. Seven-day-old offspring recovered from host mouth of (**e**) cuckoo catfish with fully filled guts and (**f**) *S. granulosus* with sparse remnants of yolk in their alimentary tracts. **g** Proportion of surviving embryos in a Lake Tanganyika (able to respond to cuckoo catfish parasitism) cichlid host (*Shuja horei*, egg diameter 4.7 mm) over the period of 4–7 days ($n = 18$ eggs across 6 clutches for *S. granulosus* and 20 eggs across 6 clutches in the cuckoo catfish controls). Bars and whiskers represent overall estimated means and one standard errors, while the individual points illustrate the proportions of surviving *Synodontis* embryos in particular cichlid clutches. Source data are provided as a Source Data file.

cuckoo catfish often consume host eggs when they intrude during host spawning[16], we revealed that they are not specialized egg predators. Occasional interactions with cichlid hosts, when feeding on their eggs, may have been sufficient to trigger the origin of obligate brood parasitism, perhaps facilitated by the exploitation of the strong maternal instinct of mouthbrooding cichlids[42]. This highlights the importance of pre-existing traits in the host, along with those in the parasite, at the origin of brood parasitism.

The evolutionary innovations of the cuckoo catfish were formed by selection on already existing traits. These traits (large eggs and rapid embryo growth) may have evolved in a different context but were later co-opted to support innovations which enhanced the success of brood parasitism after it emerged. The cuckoo catfish innovations parallel the traits exemplified by virulent brood parasites among birds and consist of frequent production of small clutches, which allow for effective utilization of host availability[43,44], the evolution of egg mimicry to facilitate egg adoption[5,45], and developmental traits that enhance the performance of catfish embryos in the host's buccal cavity[4,46] by enabling them to harm host offspring and exploit host resources[1,8,47]. Overall, our study demonstrated that, despite different ecological and life history foundations of catfish brood

parasitism, the developmental consequences of the brood parasitic strategy in the cuckoo catfish mirror the traits central to the coevolution of avian brood parasites and their hosts.

## Methods

All research methods and protocols adhered to national and institutional animal care and use guidelines, administered under permit No. CZ62760203 and approved by ethical boards of the Institute of Vertebrate Biology and the Czech Academy of Sciences (approval No. 32-2019). Fieldwork and sample export have been approved by research permits (Zambia: K-4335/18 KA/K.48/18 and individual study permits of MR, HZ, RB, MP, VB and SK), Tanzania (TAFIRI/HQ/RES.CLEARANCE/82, Costech: 2022-603-NA-2022-228). Import of samples to EU has been approved by Czech State Veterinary Administration (Zambia: SVS/2024/064505-G, Tanzania: SVS/2024/064505-G).

### Adult fish collection

Adult *Synodontis* and cichlids were collected during nine field expeditions to Zambia and Tanzania between 2019 and 2024. Fish were collected during Scuba dives using hand nets, stop nets, and baited minnow traps and brought to the surface for further processing.

Additional samples (of deepwater *S. granulosus*) were obtained from deep-water gill nets deployed by local fishermen.

Fish were typically processed within the first 2 h after collection. They were euthanized with an overdose of clove oil, and a piece of fin was stored in 96% ethanol for DNA barcoding and further genetic analyzes. The fish were measured, weighed, labeled, fixed in 6% formaldehyde, and subsequently transferred to 70% ethanol for long-term storage. A piece of muscle from the fish flank (dorsolateral side) was taken and stored in ethanol for the analysis of stable isotope ratios. Vouchers are stored at the Natural History Museum in Vienna, and collections are held at the Institute of Vertebrate Biology, Czech Academy of Sciences[48]. Morphological identifications were confirmed by DNA barcoding using the cytochrome *c* oxidase I gene[22].

## Phylogeny

A detailed ddRADseq-based phylogeny with broad geographic representation has been published elsewhere[22]. To visualize the phylogenetic background of our study species, we selected one individual for each study LT *Synodontis* species and one riverine outgroup (*Synodontis nigrita*). The tree was inferred using a maximum likelihood approach in IQ-TREE v.2.0.7 and a concatenated alignment of 6536 ddRADseq loci. We used the same nucleotide substitution model (TVM +F+R2) as in ref. 22 and 5000 ultrafast bootstrap replicates.

## Dietary data and analyses

A diet analysis was conducted on samples collected from the southern tip of the lake in Zambia, specifically at Mpulungu (Chituta Bay, Kalambo Lodge, Mutondwe Island, Mpulungu) and Ndole Bay (Chimba Rocks, Mpende Fisheries, Katete, and Cape Kachese), where all examined fish species co-occur locally. All primary sample data are georeferenced[48]. For the diet analysis, we included only fish processed within 2 h of their collection. Individuals with unidentified digested matter (i.e., with fish bait used in the traps) in their gastrointestinal tract were excluded from further analyzes.

In the laboratory, all specimens were measured for total length (TL) to the nearest 1 mm and weighed (W) to the nearest 0.01 g. Gut contents were weighed to the nearest 0.001 g and stored in glycerin. Food items were subsequently identified to the lowest practical taxon (i.e., to order, family, genus, or species) under a stereomicroscope (Nikon SMZ1000, Japan) and counted, or their counts estimated in cases of high abundance of particular prey items. The total number and biomass (to the nearest 0.1 mg) of each prey type were recorded for every fish specimen. Prey items were combined by taxon and quantified by frequency of occurrence and percentage of biomass. Proportions of diet were calculated from the contents of the gastrointestinal tracts, excluding the category "animal remains" which were decayed beyond a state to identify it. Chironomidae, Ephemeroptera, Trichoptera, Coleoptera, and Heteroptera were pooled as 'benthic insects'. All other diet categories were kept separate, as we could not unambiguously distinguish their ecological origins (i.e., benthic vs. planktonic, surface-bounded vs. free roaming). We analyzed 122 fish (30 *S. multipunctatus*, 3 *S. granulosus*, 41 *S. irsacae*, 9 *S. polli*, and 39 *S. petricola*).

To test species differences in diet composition, we used Multiple Response Permutation Procedure. We calculated the average within- and between-species differences (Bray–Curtis dissimilarities, BCDs) in diet compositions (meandist; package "vegan")[49]. Dietary specialists are predicted to have small average within-species BCDs, whereas the average within-species BCDs of dietary generalists are the same as or higher than the average between-species BCDs. We then examined the overall species effect on the diet using a Multivariate Generalized Linear Model (multivariate GLM) with a negative binomial error distribution (manyglm; package "mvabund")[50]. This approach addressed the challenge of interpreting statistical significance in distance-based multivariate comparisons, which is complicated by the confounding

effect of abundance on variance in the data. Subsequently, an ANOVA was conducted to determine the importance of specific food categories in the separation across *Synodontis* species using the Likelihood Ratio statistic (Table S1). Only four food categories (algae, sponges, zoobenthic insects, Copepoda, and fish scales) had a significant effect on the model fit (Table S1). We then employed SIMPER analysis to assess the importance of individual food categories in the pairwise contrasts among *Synodontis* species. Food categories contributing cumulatively to >80% dissimilarity within each species pair are listed in Table S2. Finally, a multivariate PCoA (Principal Coordinates Analysis) was applied to represent the BCD values in Euclidean space. Since the variables "locality", "sex", and "body size" did not significantly contribute to the multivariate GLM, only the variable "species" was included in the final analysis.

## Stable isotope data and analyses

To estimate the extent of trophic niche separation among LT *Synodontis*, we measured the stable carbon ($\delta^{13}C$) and nitrogen ($\delta^{15}N$) isotope composition of 198 *Synodontis* specimens (5 species), 62 cichlids (8 species), and 34 baseline items from various trophic levels. A piece of fish muscle (or entire individuals for the baseline items) was initially stored in ethanol and dried in the laboratory. To account for a shift in the values of $\delta13C$ and $\delta15N$ caused by preservation in ethanol[51], we collected a subsample of each species as dried samples paired with the ethanol-based sample. The reported stable isotope (SI) values were corrected using the average shift in SI values for each species separately. Samples (2 mg of dry mass) were analyzed by Iso-Analytical Ltd (Crewe, UK) using the Europa Scientific 20-20 IRMS elemental analyzer calibrated against international standards. The $\delta^{13}C$ values were corrected for lipid content following established protocols[52]. Isotope values of baseline items were corrected for trophic enrichment according to specified methods[53] and organized into informative categories (i.e., macrophyta: a combination of 4 plant species). Full details are provided in the source data[48].

The $\delta^{13}C$ and $\delta^{15}N$ values were represented as biplots. Baseline and *Synodontis* values were overlaid with stable isotope signatures of cichlids (8 species with well-defined trophic positions[12], encompassing a large isospace). This allowed us to compare the degree of overlap in trophic positions of LT *Synodontis* with the locally coexisting cichlid trophic guilds. The $\delta^{15}N$ values indicate sample positions in the food chain; $\delta^{15}N$ enrichment increases with trophic levels. The $\delta^{13}C$ values help to indicate primary producers at the start of the food chain and differentiate between littoral and pelagic environments; higher values suggest shallow (littoral) sources, while lower values indicate deeper (pelagic) habitats[30,54].

We employed permutation-based analysis of variance (PERMANOVA) to examine differences in SI signatures among *Synodontis* species using the function "pairwise.adonis" (R package pairwiseAdonis)[55], which acts as a wrapper for the "adonis" function from the R package "vegan", with 10,000 permutations. We applied the Benjamini–Hochberg procedure to account for multiple testing.

## Reproductive traits

Gonads were dissected from formalin-fixed specimens (127 females, 114 males) of *Synodontis* collected at the beginning (November 2021) and final part (March 2022) of the rainy season in the Zambian region of the lake (Mpulungu area). The fish were dissected in the laboratory, the gonads were separated, weighed to the nearest 0.01 g, immersed in 70% ethanol and processed by Histoplast (Sigma). For each gonad, 60–80 semi-serial sections were examined using hematoxylin and eosin (H&E) staining under light microscopy to detect the presence of mature oocytes and spermatozoa[56]. We identified a season-specific proportion of sexually mature individuals, namely fish with mature oocytes (indicated by the presence of at least some gametes at the two latest stages of vitellogenesis or spermiogenesis)[56,57]. We compared the

proportion of ripe gonads between the beginning and end of the rainy season for each species and sex separately, using chi-squared tests.

Investment in gonads was modeled as gonad mass (mg) relative to body size (TL, mm) for females and males separately. Gonad mass was log-transformed before analysis and modeled (Linear Model) as a function of body size (TL), species, and their interaction. When the interaction between TL and species was not significant (i.e., the slope of the relationship between gonad mass and TL was not species-specific), a simplified model (without interaction) was employed, and its fit was compared with the more complex model using ANOVA. Tukey tests were used for pairwise differences when species significantly differed in their relative investment in reproduction. Two female *S. irsacae* were excluded from the statistical analysis as major outliers (ovaries of 8.03 and 8.58 g compared to <2.7 g for all other females). Including these two females in the dataset would add stronger support to the outcome of larger ovaries in *S. irsacae*. The TL measured on this set of fish was used to compare body size across *Synodontis* species.

A comparison of egg traits (size, adhesiveness, coloration) among five LT species and three riverine species of *Synodontis* was conducted on eggs obtained following hormonal stimulation of fish (Ovaprim, a salmon gonadotropin-releasing hormone analog, Syndel Laboratories Ltd, Canada)[58]. Females, under clove oil anesthesia, were injected with 0.5 µL/g of Ovaprim, and ripe eggs were gently stripped out after 12 to 24 h onto glass Petri dishes. After 3 to 5 min, the dishes containing the eggs were carefully transferred into a larger glass tank filled with water and positioned vertically (at 90°). Egg adhesion was determined as the proportion of eggs remaining on the glass surface after 5 s compared to the total number of eggs initially placed on the dish. Eggs that did not adhere to the glass surface fell away within this 5-s period. Egg adhesion was modeled as the proportion of eggs adhered to the Petri dish in a vertical position relative to the total number of eggs on the dish, using a GLM with ordered beta distribution and a logit link. The egg adhesion of the cuckoo catfish was set as the reference value, and contrasts with the egg adhesion of other species were tested.

To compare egg coloration, a subsample of eggs from each female was placed on a separate glass Petri dish. Standardized pictures (RAW format) were taken over a standard grey card (18% standard reflectance, KELDAN grey card including a scale tape) from each subsample of eggs immediately after the test for egg adhesiveness. Raw pictures were adjusted to the grey background in the Mica ToolBox (ImageJ plugin[59], and converted into a linear multispectral image. The linear images were transformed into cone-catch images (representing the human visible range in daylight) by using a custom-prepared cone-catch model for the digital camera used (Canon 7D). This was carried out by taking a RAW image of the KELDAN color checker under natural daylight, preparing a linear multispectral image of the color checker, and following the standard protocol and the built-in algorithm of the Mica ToolBox to generate a cone mapping model (using the CIE 1931 400–700 nm, D65 400–700 nm configuration). The cone-catch images were then further transformed into CIELab color space within the Mica ToolBox. Seven to ten eggs were measured per individual clutch (except for only three eggs in a *S. macrostigma* clutch). The average values for each clutch were used in further analysis. Egg coloration was quantified by CIE_A and CIE_B axes. The CIE_A represents the red-green continuum (positive values: red, negative values: green, zero: grey). The CIE_B represents the yellow-blue continuum (positive values: yellow, negative values: blue, zero: grey). A Gaussian Linear Model was used to compare the CIE_B and CIE_A among species, with the cuckoo catfish set as the reference species. The contrasts between the cuckoo catfish and the other species are reported.

To analyze egg size, the images taken for color comparison (which included a scale) were used, and the egg diameters were measured using ImageJ 1.53t. A Linear Mixed Model (LMM) with Gaussian error distribution compared egg sizes across species. Up to 10 eggs were measured from the same clutch, with clutch ID modeled as a random intercept to account for the lack of independence. The cuckoo catfish served as the reference species.

Clutch size was estimated from the same set of clutches as other traits (i.e., following hormonal stimulation). The eggs were counted on Petri dishes. A linear model using log-transformed data was employed to calculate interspecific contrasts. We also compared the clutch sizes obtained during natural egg maturation and following hormonal stimulation in the cuckoo catfish (where comparison was feasible), finding no difference (Linear Model, $P = 0.728$, $N = 9$ and 10 clutches for hormonally induced and natural clutches, respectively). All statistical analyzes were conducted using *glmmTMB*[60]. We tested model dispersion, zero inflation, and model misspecification using the *DHARMa* package[61]. Post-hoc pairwise comparisons were calculated using *emmeans*[62] with Benjamini–Hochberg procedure for multiple testing.

We then used the *phytools* package[63] to reconstruct ancestral character states and the *geiger* package[64] to calculate evolutionary signal metrics. Blomberg's K measures the relative strength of phylogenetic signal, estimating how closely related species resemble each other compared with expectations under Brownian Motion (BM) evolution. We used Pagel's λ to estimate the extent to which the phylogeny explains trait covariance. *Synodontis nigriventris* and *S. macrostigma* were used as outgroups.

### Developmental traits

To compare the growth and development of LT *Synodontis*, we conducted in vitro fertilization in the laboratory to raise the offspring under standardized conditions. Wild-caught adult fish were treated with Ovaprim, according to ref. 58, with species-specific modifications in timing. The eggs and sperm were extracted from the adult fish and mixed for a minimum of 10 min. The fertilized eggs were placed in a water recirculation system and maintained at $27 \pm 0.5$ °C. One day post-fertilization, all non-developing (i.e., unfertilized) eggs were removed. Embryo samples were collected at 24-h intervals until the age of 10 days post fertilization and stored in 4% PFA at 4 °C.

The body size of fish was measured from the snout to the end of the caudal fin (Total Length, TL) to the nearest 0.1 mm. Growth rate was modeled using a Gaussian Linear Model on log-transformed TL, with species, age, and their interaction as fixed factors in the *glmmTMB* package. Species-specific contrasts were also analyzed at 10 dpf.

To assess the development of dentition, Paraformaldehyde-stored embryos (PFA buffer, 4%) of all LT and two outgroup *Synodontis* species were washed in PBS for 10 min at room temperature (RT) and then bleached in a solution containing 1% KOH and 3% hydrogen peroxide (1:1) under light. Subsequently, the specimens were washed in distilled water for 30 min at RT and placed in a saturated sodium tetraborate (Sigma-Aldrich: cat. No. 71996) solution overnight. Further staining for mineralized tissues was carried out in Alizarin Red S (Thermo Scientific Chemicals, cat. No. 400481000) solution in 1% KOH overnight, followed by washing in distilled water for 30 min. Embryos were then transferred to 1% trypsin (Sigma-Aldrich, cat. No. T4799) in 2% sodium tetraborate solution for 1–3 days (depending on the embryo size). After that, specimens were moved into glycerol through a series of glycerol in 1% KOH solution (25%, 50%, 75%) with each step performed overnight. Once in glycerol, the specimens were squeezed and coverslipped. Photographs were taken using an Olympus BX51 microscope under fluorescent light with an Olympus DP74 camera, employing the cellSens software. Final images were composed in Photoshop 2023 from photos of varying focus distances; contrast and colors were adjusted.

For skeletal development, staining followed the protocol by ref. 65. Photographs were taken using an Olympus SZX12 stereoscopic microscope equipped with an Olympus u-tv0.5xc-3 camera and QuickPhoto Micro (ver. 2.3) software, during which Z-stacking was also

performed. Final images were edited in Photoshop to smooth the background and minimize distraction.

## Potential for *S. granulosus* parasitism

Based on the reproductive and developmental traits of *S. granulosus*, we experimentally assessed whether the existing set of traits enables its offspring to survive in the buccal cavity of a mouthbrooding cichlid. Six pairs of wild-caught adult *S. granulosus* were hormonally stimulated (Ovaprim, as detailed above) for in vitro fertilization to obtain viable embryos. The fertilized eggs were then placed in tumblers with recirculating incubation water[27]. Water for incubation was prepared using reverse osmosis water with 11 g/L Cichlid Lake Salts (Seachem Laboratories Inc., USA). The pH was adjusted to 8.5–9 using Tanganyika Buffer (Seachem Laboratories Inc., USA), and the temperature was set to 27 (±0.5) °C. The eggs of *S. multipunctatus* (used as parasitic controls) were obtained from 10 pairs and treated in the same manner as the eggs of *S. granulosus*. The eggs of non-parasitic control *S. polli* were obtained from 9 pairs of hormonally stimulated parents. As the experiments with *S. granulosus* and *S. polli* were conducted at different time, a separate control using *S. multipunctatus* eggs (obtained from 10 parental pairs) was completed during the *S. polli* experiment. After 24 h of incubation, the experimental eggs were examined under a binocular microscope to confirm their proper morphological development. The same water quality was used for incubation and subsequent experiments.

To maximize the simultaneous availability of freshly spawned mouthbrooding females, we separated the sexes of both host species (*S. granulosus* experiment: 54 females and 19 males of *Haplochromis* sp. CH44; 30 females and 15 males of *S. horei* for 2 weeks; *S. polli* experiment: 69 females and 16 males of *Haplochromis* sp. CH44 for 3 weeks). Subsequently, we housed females and males together for 3 days and used any freshly mouthbrooding cichlid female from the two species for inoculation. *Haplochromis* sp. CH44 is a species from Lake Victoria, known to be susceptible to experimental brood parasitism by *S. multipunctatus* eggs and embryos[13] and producing relatively small eggs (2.5 mm). *Shuja horei* is a natural host of *S. multipunctatus*, which likely possesses defense mechanisms[40] and produces larger eggs (4.7 mm). We used two or four developing *Synodontis* eggs at 24 h post fertilization to artificially inoculate a mouthbrooding female. The eggs were collected into a Pasteur pipette, which was then gently inserted into the mouths of the brooding cichlid female and released[13].

Inoculated host females were transferred to 60 L housing tanks, which were equipped with a perforated false bottom (mesh size 0.5 cm), an air-driven sponge filter, and a clay pot serving as a shelter. The tanks were dimmed to minimize external disturbances. The females were allowed to incubate their clutches for 4 days (*S. horei* host: 3 clutches per *Synodontis* species; *Haplochromis* host: 2 clutches of *S. granulosus*) and 7 days (*S. horei* host: 3 clutches per *Synodontis* species; *Haplochromis* host in *S. granulosus* experiment: 4 clutches of *S. granulosus* and 7 clutches of *S. multipunctatus*; *Haplochromis* host in *S. polli* experiment: 5 clutches of *S. polli* and 7 clutches of *S. multipunctatus*). Because *S. polli* embryos had zero survival in the benign *Haplochromis* hosts, we did not inoculate their eggs into more challenging host (Tanganyikan *S. horei*). In contrast, the high survival rate of *S. granulosus* in *Haplochromis* hosts led us to extend the inoculation of 3 clutches to 12 days to test whether the offspring of *S. granulosus* can survive in the host's buccal cavity after all resources from their own yolk sac were depleted (which occurs at day 6 post fertilization) and their survival depended on extracting resources from the host clutch. Finally, we further extended the duration of incubation to 14 days, which covers the period of host offspring release. At the end of incubation (day 4, 7 or 12 post fertilization), mouthbrooded clutches were flushed from the female's mouth using a Pasteur pipette, and surviving

*Synodontis* offspring were counted. For replicates with 14 days post fertilization, all live offspring retained in the tank (protected under the false perforated bottom) were counted and scored as surviving.

The success or failure of development of *S. granulosus* in cichlid buccal cavities was tested using GLMM with binomial error distribution, with *Synodontis* species (*S. granulosus* as treatment, *S. multipunctatus* as control), duration of incubation, and their interaction (to account for potential species-specific survival) as fixed factors. The interaction term was not significant ($P = 0.635$) and was removed from the final model. The host female identity was used as a random intercept to account for non-independence of groups of 2 or 4 *Synodontis* eggs incubated in the same host. The analyzes of experiments with *S. granulosus* and *Shuja horei* and with *S. polli* and *Haplochomis* did not converge, likely due to zero survival in the treatment groups (0 of 18 *S. granulosus* eggs compared to 6 or 20 control cuckoo catfish eggs; and 0 of 20 *S. polli* eggs compared to 26 of 28 control cuckoo catfish eggs, respectively). While Fisher's exact test demonstrated significantly lower survival of *S. granulosus* in *S. horei* ($P = 0.021$) and of *S. polli* in *Haplochromis* ($P < 0.0001$), we do not report these analyzes in the main text as the input data were more structured (grouped by host female individuals) than Fisher's exact test can handle. Instead, we declare zero survival of treatment embryos in mouthbrooding hosts which strongly contrasts with common survival of their respective controls.

All statistical tests were two-tailed. All individual measurements were taken from distinct individuals. Any covariates dropped from the final models are reported in the methods.

## Reporting summary

Further information on research design is available in the Nature Portfolio Reporting Summary linked to this article.

## Data availability

All trophic, reproductive, ontogenetic and experimental data generated in this study have been deposited in the Figshare database [https://doi.org/10.6084/m9.figshare.28646141] and are freely available. Source data are provided as a Source Data file. Raw ddRADseq reads have been deposited in NCBI's Sequence Read Archive (https://www.ncbi.nlm.nih.gov/sra) (BioProject: PRJNA1132910) under accession numbers SAMN42360897–SAMN42360967. Source data are provided with this paper.

## Code availability

The scripts to replicate all analyses reported in this study are available from Figshare [https://doi.org/10.6084/m9.figshare.28646141].

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

## Acknowledgements

We thank the Department of Fisheries in Mpulungu, especially Lwabanya Mabo, for facilitating research in Zambia, and Tanzanian Fisheries Research Institute in Kigoma, especially Deogratias P. Mulokozi, for facilitating research in Tanzania. The manuscript benefited from comments from Carl Smith, Lukáš Kratochvíl, Miranda Sherlock and four anonymous referees. This research was funded by the Czech Science Foundation (21-00788X to M.R.) and, in part, by the Austrian Science Fund (FWF) (doi.org/10.55776/J4584 to H.Z). For the purpose of open access, the authors have applied a CC BY public copyright license to any Author Accepted Manuscript version arising from this submission.

## Author contributions

Conceptualization: M.R., S.K., H.Z. Methodology: M.R., H.Z., R.B., K.P., J.Ž., R.C. Investigation: M.R., H.Z., R.B., T.S., M.P., G.E., V.B., K.P., J.Ž., L.K., I.D., S.K. Visualization: M.R., H.Z., T.S., G.E. Funding acquisition: M.R., H.Z. Project administration: M.R. Supervision: M.R. Writing – original draft: M.R., H.Z. Writing – review and editing: M.R, H.Z., R.B., T.S., M.P., G.E., V.B., K.P., J.Ž., L.K., I.D., R.C., S.K.

## Competing interests

The authors declare no competing interests.
