## [Transparent Peer Review file · Nature Communications]

The ecological and developmental foundations of brood parasitism in a catfish

Corresponding Author: Professor Martin Reichard

Version 0:

Reviewer comments:

Reviewer #1

(Remarks to the Author)

The manuscript by Reichard and colleagues describes an investigation of the evolution of brood parasitism in a species of African Rift Lake catfish, *Synodontis multipunctatus*, that manipulates mouth-brooding cichlids into caring for its young, with the catfish larvae additionally feed on the host's own offspring. As the authors point out, their system involves the evolution of brood parasitism in a lineage with an ancestral condition of no parental care. This situation contrasts with more extensively studied examples in birds, where the ancestral condition was the presence of parental care.

The authors first test the hypothesis that the evolution of brood parasitism in *S. multipunctatus* was facilitated by an initial stage of adults feeding on the cichlid eggs; such trophic interactions have been documented in other systems of brood parasitism. They document the gut contents of wild-caught specimens of five species of Lake Tanganyika *Synodontis*, finding variation in diet among species and that *S. multipunctatus* has a generalized diet, rather than being a specialized egg predator. This result was further supported by stable isotope analysis of muscle tissue from the *Synodontis* species.

The authors next induced reproduction in the lab of the same five Lake Tanganyika *Synodontis* species along with three riverine outgroups in the genus. They compared reproductive characteristics, as well as embryonic and larval development, to determine whether several proposed adaptations to brood parasitism arose before or after the divergence of *S. multipunctatus* from its egg-scattering relatives. Among the reproductive characteristics of *S. multipunctatus*, large egg size, low adhesiveness of eggs, and lack of spawning seasonality appeared to have been present before the evolution of brood parasitism. In contrast, egg color mimicking that of the host cichlids as well as the frequent production of small clutches of eggs appear to be unique to *S. multipunctatus*.

Similar analysis of embryonic and larval development suggested that the rapid larval growth of *S. multipunctatus* is shared with its sister species *S. granulatus*, while rapid mineralization of oral and pharyngeal teeth and jaws (enabling feeding on cichlid eggs) is unique to the former species.

Finally, the authors tested whether larvae of *S. granulatus* could survive in the mouths of cichlids and/or feed on the young of the host. They found that they could survive, but their ability to feed on eggs was significantly lower than that of *S. multipunctatus*. Such survival was also greater in a cichlid from a different lake compared to that in a sympatric cichlid that has likely evolved defenses against parasitism by *S. multipunctatus*.

This is a well-designed and executed study, and the manuscript is clearly written. The *S. multipunctatus* system is a fascinating one that I believe will be of wide interest. The finding that some of the likely adaptations for brood parasitism were present before its evolution is also likely to be of general interest to evolutionary biologists

I have only minor suggestions for improvement as follows.

1) Evolution of reproductive parasitism from species with no parental care brought to my mind the "nest associates" that spawn in the pebble mounds of the North American chubs of the genus *Nocomis*. (Betts et al 2024. *Freshwater Biology*, 69, 450–459. <https://doi.org/10.1111/fwb.142242>). Is this of any relevance to the *Synodontis* system? If not, there is no need to comment on it in the manuscript.

- 2) The authors find that the diet of *S. granulosus* consisted exclusively of fish tissue. Can they speculate on whether this is the ancestral condition of the *S. multipunctatus* lineage, and if so, whether this might be the trophic interaction between catfishes and cichlids that preceded brood parasitism?
- 3) It appears from Figure 2 (e and f) that the larger egg size of *S. multipunctatus* is not associated with larger adult body size. Can the authors comment explicitly on the adult size of the five Lake Tanganyika species?
- 4) In line 224, it is indicated that host offspring disappeared in the presence of inoculated *S. granulosus*. Was this disappearance complete in both *S. multipunctatus* and *S. granulosus*, or was it less in the latter species?
- 5) In lines 254-255 the authors suggest that “Occasional interactions with cichlid hosts may have been sufficient to trigger the origin of obligate brood parasitism ...” Can they speculate on what these interactions might be? Do the cichlids occupy areas that would be attractive for egg scattering? Is the spatial overlap between *S. multipunctatus* and the host cichlids known when the former is not spawning?

Reviewer #2

(Remarks to the Author)

The manuscript “The ecological and developmental foundations of brood parasitism in a catfish” presents a comprehensive and well-written study that aims to shed light on the origin and early evolution of brood parasitism within the Lake Tanganyika Synodontis radiation. It seems this system is particularly intriguing because it offers a rare opportunity to study the evolution of brood parasitism in a lineage that lacks parental care.

The authors adopt an integrative approach, which I find highly commendable, combining multivariate ecological quantification, various reproductive life-history traits, developmental series (covering growth, dental development rates), and an experimental brood parasitism assay. The scope of the data collection is impressive and I really enjoyed reading the manuscript!

In summary, the main findings indicate that the cuckoo catfish differs from its closest relatives in numerous traits, particularly in egg coloration, clutch size, egg size, and the timing of jaw development. Furthermore, both the focal species and its sister species exhibit elevated growth rates and reduced egg adhesion. Taken together, the suite of traits observed in the focal species align well with characteristics expected to facilitate brood parasitism.

The breadth of data surely provides valuable context for discussing the evolutionary foundations of brood parasitism from multiple perspectives, which I consider a major strength of the work. However, I have a few concerns regarding the extent to which the analyses fully address the central evolutionary question, as well as some minor specific comments:

1. The conclusive power of the model system

All statistical tests in the manuscript address whether the cuckoo catfish *S. multipunctatus* differs from other species. While valid in that context, this does not answer the broader question implied by the authors: what traits relate to the origin and evolution of brood parasitism? Addressing that question would require phylogenetic comparative analyses – which are absent from the manuscript.

Unfortunately, the nature of the system offers very limited statistical power, as it effectively has a comparative sample size of $N = 1$ for brood parasitism (since the trait appears to have evolved only once within the clade). Given this limitation, can we realistically identify traits that are statistically linked to the origin and evolution of brood parasitism?

Undoubtedly, the observed trait shifts make intuitive sense in light of brood parasitism and generate intriguing hypotheses; nevertheless, the work remains primarily descriptive and hypothesis-generating, rather than formally detecting signals of adaptation. While this work is a valuable contribution – especially for broader comparative context or discussions across systems – the limitations should be explicitly acknowledged, and a more cautious tone adopted when interpreting patterns as adaptive responses or drivers.

2. The experimental brood parasitism assay

This experiment is highly intriguing and a valuable step toward understanding potential parasitism in *S. granulosus*.

However, its interpretative strength could be improved by including a known substrate-spawning Synodontis as a negative control.

The current experimental design only demonstrates that *S. granulosus* offspring do not survive in what they call “sympatric host” mouths. However, the absence of a non-parasitic control makes it difficult to assess whether the high survival in the naïve host (where both species performed well) is expected for a non-parasitic Synodontis or if *S. granulosus* falls somehow along a parasitic spectrum?

A second concern I have is the match between *S. granulosus* and the tested host species *S. horei*. As mentioned in the manuscript, *S. granulosus* is a deep-water species (>40 m), whereas *S. horei* is a well-documented shallow-water dweller rarely encountered below 5–10 m and is hence unlikely a true sympatric host for *S. granulosus* (the match with no embryo survival). In contrast, *S. horei* is a known true sympatric host of *S. multipunctatus* (the match that performed very well). So, how can host-specificity within the lake be excluded as a driver of this pattern?

Minor specific comments:

- The use of the term Pre-adaptation: This may seem like a matter of semantics, but I would encourage the authors to reflect on their use of the term pre-adaptation. While it has been widely used historically, the term can imply evolutionary foresight. The perhaps more widely accepted term is exaptation, which conveys that a trait originally evolved for one function (or without a specific function) and was later co-opted for another. Regardless of which term is chosen, I recommend briefly defining it at first use to ensure clarity.

- L- 63: In the introduction, the hypothesis: "We hypothesized that the origin of brood parasitism in the cuckoo catfish has primarily arisen from the predation of adult catfish on cichlid eggs during spawning" does feel somewhat abrupt since there is little lead-up to the idea that predation might be the key precursor.

- Figure 1: The color coding of *Synodontis* species is inconsistent across panels (e.g., in Fig. 1a *S. granulosus* is dark blue, whereas in Fig. 1b it appears purple)?

- L.93: It is mentioned that fish eggs were found in stomach contents, but I cannot locate them in the overview of dietary items shown in Fig. 1b. If they are included under a broader category, please clarify this in the text or figure legend.

- Figure 1b: Cladocera are listed as prey items. To my knowledge, water fleas do not occur in Lake Tanganyika. Could this be a taxonomic mix-up, or is there a specific explanation for their presence? Clarifying this would be helpful.

- L.93–94: The statement "However, the small size and spherical shape of consumed eggs clearly indicated the eggs did not belong to a mouthbrooding cichlid species." may be a bit too general. For instance, the mouthbrooder *Perissodus microlepis* has very small, nearly spherical eggs. Perhaps clarify that they were evidently not from the major known host species rather than ruling out all mouthbrooders?

- L.100–101: How was it determined whether the bones or scales belonged to cichlid or non-cichlid species?

Version 1:

Reviewer comments:

Reviewer #1

(Remarks to the Author)

The revised version of the manuscript addresses all of the concerns I raised in my previous review in a thorough and satisfactory manner.

Reviewer #2

(Remarks to the Author)

The authors have now thoroughly revised the manuscript and have addressed all of my previous comments. In particular, they have added phylogenetic comparative perspectives, included a negative control in the parasitism assay, and clarified all minor points raised.

These additional analyses represent a valuable extension to the manuscript. Specifically, the ancestral state reconstructions now formally document the relative timing of trait evolution, which in the previous version was only visually inferred.

Likewise, the inclusion of *S. polli* as a negative control has clearly strengthened the experimental design. Together, these changes have substantially improved the manuscript.

That said, I still have two minor follow-up comments, as outlined below.

1) ancestral trait reconstructions

I am not sure whether I agree with the statement that these particular analyses "suffer" from the single evolutionary origin of brood parasitism (e.g. L173–174: "Despite the interpretative limitation imposed by a single evolutionary origin of brood parasitism, significant ...").

My original concern about the effective comparative sample size of $N = 1$ applies specifically to formal tests of correlated evolution between traits and parasitism. In other words, the limitation imposed by a single origin of brood parasitism constrains analyses that would ask whether parasitic versus non-parasitic lineages differ in trait values once shared ancestry is accounted for (e.g. phylogenetic ANOVA or PGLS with brood parasitism as a predictor). Such analyses are not conducted here – instead the focus is on differences between species.

While I think the manuscript is in general carefully worded not to oversell it as a test for evolutionary correlations, there are (still) instances where it is implied. For example, in the reporting of statistical results in Fig. 2 the caption states: "Asterisks indicate whether clutches of the cuckoo catfish were significantly smaller than those of all other species (LM, $P < 0.001$)." If the question is whether *S. multipunctatus* differs from other species individually, we expect pairwise comparisons not one statistical test. However, if the question is whether parasitic species differ from non-parasitic species as a group, then a comparative model accounting for phylogenetic non-independence would be required. I therefore recommend revisiting the wording throughout the manuscript to ensure that this distinction is consistently maintained and that statistical tests are described in a way that matches the questions they address.

2) experimental parasitism

Despite the fact that *S. granulosus* performs more similarly to the cuckoo catfish than to the negative control in a naïve host,

the interpretation of these results does not seem to have been fully adjusted. In light of these findings, one of my questions from the previous round remains: how can host specificity within the lake be excluded as a driver of the observed patterns? Host–parasite interactions and adaptations go both ways. The authors argue that *S. horii* has evolved defence mechanisms through coevolution with *S. multipunctatus*. It seems equally plausible, however, that *S. multipunctatus* is specialized on its natural shallow-water hosts (shallow-water haplochromines). We do not know whether *S. multipunctatus* would perform equally well in a deep-water, non-haplochromine mouthbrooder — a “challenging” but potentially not compatible host. Basically, the reciprocal scenario relevant if *S. granulosus* were a brood parasite of deep-water hosts. Relatedly, several arguments used to reject brood parasitism in *S. granulosus* appear to implicitly take shallow-water haplochromines as the relevant reference. For example, statements that its eggs do not visually imitate host eggs overlook the considerable variation in egg size, colour, and clutch structure among mouthbrooding cichlids. Adaptation to different hosts could plausibly result in different egg phenotypes. Similarly, arguments based on large clutch size do not fully exclude alternative scenarios, such as egg distribution across multiple hosts e.g. during lek-like spawning events. Having said that, I want to emphasize that I am not arguing that *S. granulosus* is necessarily a brood parasite. Rather, I think the current results make it difficult to rule out host-specific parasitic strategies and suggest that traits that are expressed differently can still be relevant to brood parasitism. I therefore encourage the authors to discuss the evidence for and against parasitism in *S. granulosus* in a more balanced way (e.g. L 264-273).

We thank both reviewers for their thoughtful and constructive feedback, which has substantially improved our manuscript. We carefully addressed every comment and revised the text for clarity, accuracy, and consistency. This includes refined interpretations, corrected terminology and clarified diet classifications. Most notably, in response to the reviewers' suggestions, we performed a new artificial parasitism experiment using a non-parasitic *Synodontis* species, which strengthened our key conclusions. We also conducted ancestral trait reconstruction as suggested by the reviewer 2. Accordingly, we added new supporting information, updated data visualizations and files submitted to Figshare repository.

We hope that these revisions, new analysis, and additional experiment fully addressed the concerns raised by the reviewers. Our detailed responses to each point follows.

Finally, we also added a short summary of results at the end of introduction, as requested for Nature Communications paper (our paper has been transferred from Nature submission which does not include that paragraph).

RESPONSE TO REVIEWERS' COMMENTS

Reviewer #1 (Remarks to the Author):

The manuscript by Reichard and colleagues describes an investigation of the evolution of brood parasitism in a species of African Rift Lake catfish, *Synodontis multipunctatus*, that manipulates mouth-brooding cichlids into caring for its young, with the catfish larvae additionally feed on the host's own offspring. As the authors point out, their system involves the evolution of brood parasitism in a lineage with an ancestral condition of no parental care. This situation contrasts with more extensively studied examples in birds, where the ancestral condition was the presence of parental care.

The authors first test the hypothesis that the evolution of brood parasitism in *S. multipunctatus* was facilitated by an initial stage of adults feeding on the cichlid eggs; such trophic interactions have been documented in other systems of brood parasitism. They document the gut contents of wild-caught specimens of five species of Lake Tanganyika *Synodontis*, finding variation in diet among species and that *S. multipunctatus* has a generalized diet, rather than being a specialized egg predator. This result was further supported by stable isotope analysis of muscle tissue from the *Synodontis* species.

The authors next induced reproduction in the lab of the same five Lake Tanganyika *Synodontis* species along with three riverine outgroups in the genus. They compared reproductive characteristics, as well as embryonic and larval development, to determine whether several proposed adaptations to brood parasitism arose before or after the divergence of *S. multipunctatus* from its egg-scattering relatives. Among the reproductive characteristics of *S. multipunctatus*, large egg size, low adhesiveness of eggs, and lack of spawning seasonality appeared to have been present before the evolution of brood parasitism. In contrast, egg color mimicking that of the host cichlids as well as the frequent production of small clutches of eggs appear to be unique to *S. multipunctatus*.

Similar analysis of embryonic and larval development suggested that the rapid larval growth of *S. multipunctatus* is shared with its sister species *S. granulosus*, while rapid mineralization of oral and pharyngeal teeth and jaws (enabling feeding on cichlid eggs) is unique to the former species.

Finally, the authors tested whether larvae of *S. granulosus* could survive in the mouths of cichlids and/or feed on the young of the host. They found that they could survive, but their ability to feed on eggs was significantly lower than that of *S. multipunctatus*. Such survival was also greater in a cichlid from a different lake compared to that in a sympatric cichlid that has likely evolved defenses against parasitism by *S. multipunctatus*.

This is a well-designed and executed study, and the manuscript is clearly written. The *S. multipunctatus* system is a fascinating one that I believe will be of wide interest. The finding that some of the likely adaptations for brood parasitism were present before its evolution is also likely to be of general interest to evolutionary biologists.

RESPONSE: Thank you very much for positive feedback and a nice summary of our study.

I have only minor suggestions for improvement as follows.

1) Evolution of reproductive parasitism from species with no parental care brought to my mind the “nest associates” that spawn in the pebble mounds of the North American chubs of the genus *Nocomis*. (Betts et al 2024. *Freshwater Biology*, 69, 450–459. <https://doi.org/10.1111/fwb.142242>). Is this of any relevance to the *Synodontis* system? If not, there is no need to comment on it in the manuscript.

RESPONSE: This system appears not to be directly relevant to *Synodontis* brood parasitism, unless we address this issue in a broader context. Several instances of brood parasitism are known in fishes, although the terminology distinguishing “nest association” from “brood parasitism” remains debated. The topic is too broad to fall into the scope of the present study. Actually, we prepare a separate opinion paper that summarizes and reviews all documented cases of brood parasitism in fishes, including a discussion of which systems should (and should not) be classified as true brood parasitism. Established brood parasites include the lineage of *Pungtungia* and *Pseudopungtungia* minnows in East Asia, as well as recently identified cases of brood parasitic catfish (other than the cuckoo catfish) from Lake Malawi and Lake Tanganyika. The North American minnows that act as nest associates are particularly difficult to categorize, as both costs and benefits may accrue to the “host” species. Given this complexity and to avoid digressing from the focus of our current study, we prefer not to include this discussion in the manuscript.

2) The authors find that the diet of *S. granulosus* consisted exclusively of fish tissue. Can they speculate on whether this is the ancestral condition of the *S. multipunctatus* lineage, and if so, whether this might be the trophic interaction between catfishes and cichlids that preceded brood parasitism?

RESPONSE: We emphasize that the feeding of *S. granulosus* likely reflects scavenging or scale-eating rather than active predation. We have clarified its classification as a fish scavenger when discussing the stable isotope results and, for consistency, now also briefly mention other *Synodontis* species. Although it might be possible to speculate about a link between ancestral state of consumption of (non-egg) fish tissue and the evolution of brood parasitism, we believe that any such association remains highly speculative and our data cannot illuminate the issue.

3) It appears from Figure 2 (e and f) that the larger egg size of *S. multipunctatus* is not associated with larger adult body size. Can the authors comment explicitly on the adult size of the five Lake Tanganyika species?

RESPONSE: Thank you for this comment. *S. multipunctatus* is indeed one of the smaller *Synodontis* species in Lake Tanganyika. We have now added explicit body-size

information of all LT *Synodontis* species (mean and s.e. of the specimens used to estimate gonadal investment) to the main text, in addition to the information that could previously be inferred from Fig. 2e and 2f (now 2c and 2d).

4) In line 224, it is indicated that host offspring disappeared in the presence of inoculated *S. granulatus*. Was this disappearance complete in both *S. multipunctatus* and *S. granulatus*, or was it less in the latter species?

RESPONSE: Yes, survival of host offspring was zero in both *S. multipunctatus* and *S. granulatus*. We have clarified this in the text by stating that no host embryos were present in experimentally parasitized females at 12–14 days post-fertilization (for *S. multipunctatus*, n = 13; for *S. granulatus*, n = 9).

5) In lines 254-255 the authors suggest that “Occasional interactions with cichlid hosts may have been sufficient to trigger the origin of obligate brood parasitism ...” Can they speculate on what these interactions might be? Do the cichlids occupy areas that would be attractive for egg scattering? Is the spatial overlap between *S. multipunctatus* and the host cichlids known when the former is not spawning?

RESPONSE: We now clarify that the initial interactions were most likely related to egg predation and have added the phrase “when feeding on their eggs.” Host cichlids and cuckoo catfish broadly co-occur in rocky littoral habitats of Lake Tanganyika, and both are among the most common fishes in these areas. Consequently, spatial overlap and frequent encounters occur even outside spawning events, making initially opportunistic interactions (such as egg predation) a plausible precursor to the evolution of obligate brood parasitism.

Reviewer #2 (Remarks to the Author):

The manuscript “The ecological and developmental foundations of brood parasitism in a catfish” presents a comprehensive and well-written study that aims to shed light on the origin and early evolution of brood parasitism within the Lake Tanganyika *Synodontis* radiation. It seems this system is particularly intriguing because it offers a rare opportunity to study the evolution of brood parasitism in a lineage that lacks parental care.

The authors adopt an integrative approach, which I find highly commendable, combining multivariate ecological quantification, various reproductive life-history traits, developmental series (covering growth, dental development rates), and an experimental brood parasitism assay. The scope of the data collection is impressive and I really enjoyed reading the manuscript!

In summary, the main findings indicate that the cuckoo catfish differs from its closest relatives in numerous traits, particularly in egg coloration, clutch size, egg size, and the timing of jaw development. Furthermore, both the focal species and its sister species exhibit elevated growth rates and reduced egg adhesion. Taken together, the suite of traits observed in the focal species align well with characteristics expected to facilitate brood parasitism.

The breadth of data surely provides valuable context for discussing the evolutionary foundations of brood parasitism from multiple perspectives, which I consider a major strength of the work. However, I have a few concerns regarding the extent to which the analyses fully address the central evolutionary question, as well as some minor specific comments:

1. The conclusive power of the model system

All statistical tests in the manuscript address whether the cuckoo catfish *S. multipunctatus* differs from other species. While valid in that context, this does not answer the broader question implied by the authors: what traits relate to the origin and evolution of brood parasitism? Addressing that question would require phylogenetic comparative analyses – which are absent from the manuscript.

Unfortunately, the nature of the system offers very limited statistical power, as it effectively has a comparative sample size of $N = 1$ for brood parasitism (since the trait appears to have evolved only once within the clade). Given this limitation, can we realistically identify traits that are statistically linked to the origin and evolution of brood parasitism?

Undoubtedly, the observed trait shifts make intuitive sense in light of brood parasitism and generate intriguing hypotheses; nevertheless, the work remains primarily descriptive and hypothesis-generating, rather than formally detecting signals of adaptation. While this work is a valuable contribution – especially for broader comparative context or discussions across systems – the limitations should be explicitly acknowledged, and a more cautious tone adopted when interpreting patterns as adaptive responses or drivers.

RESPONSE: We agree that the single origin of brood parasitism in our study system imposes inherent limitations on statistical inference. To address this in the revised version, we have conducted ancestral trait reconstructions and calculated evolutionary signal metrics (Pagel's λ and Blomberg's K) for traits related to reproduction. Given the small comparative sample size, we recognize that these analyses have limited power. Accordingly, we present them outside the main ms (Extended Data. Fig. 1) and interpret them cautiously.

More broadly, we explicitly acknowledge in the revised manuscript that the “single evolutionary origin (of brood parasitism in this clade) without a replication” necessarily restricts comparative power. As a result, our study cannot provide conclusive tests of correlated evolution. We highlight this limitation clearly (i.e., single origin) and have adjusted the tone of our interpretations to reflect this limitation.

2. The experimental brood parasitism assay

This experiment is highly intriguing and a valuable step toward understanding potential parasitism in *S. granulosus*. However, its interpretative strength could be improved by including a known substrate-spawning *Synodontis* as a negative control.

The current experimental design only demonstrates that *S. granulosus* offspring do not survive in what they call “sympatric host” mouths. However, the absence of a non-parasitic control makes it difficult to assess whether the high survival in the naïve host (where both species performed well) is expected for a non-parasitic *Synodontis* or if *S. granulosus* falls somehow along a parasitic spectrum?

RESPONSE: We fully agree with the reviewer's suggestion and have conducted an additional experiment using *S. polli* as a non-parasitic, substrate-breeding control. To control for potential effects of age or condition of a new host cichlid cohort used in this experiment, we included a new cuckoo catfish control in the experiment. The results were clear: no *S. polli* embryos survived after 7 days of incubation in allopatric *Haplochromis* hosts, whereas cuckoo catfish embryos survived successfully (93%) over the same period, consistent with prior experiments.

We have elaborated on this experiment in the main text and updated the Methods accordingly. Figure 4 has been modified to include the new dataset (panel with *S. polli* results) and presenting now results for *S. granulosus* and cuckoo catfish embryos separately for 4–7 days and 12–14 days periods (different panels). This allows direct visual comparison of embryo survival between parasitic and non-parasitic *Synodontis* for specific developmental periods. The previous version had combined data where time periods were separated by distinctive data point symbols.

A second concern I have is the match between *S. granulosus* and the tested host species *S. horei*. As mentioned in the manuscript, *S. granulosus* is a deep-water species (>40 m), whereas *S. horei* is a well-documented shallow-water dweller rarely encountered below 5–10 m and is hence unlikely a true sympatric host for *S. granulosus* (the match with no embryo survival). In contrast, *S. horei* is a known true sympatric host of *S. multipunctatus* (the match that performed very well). So, how can host-specificity within the lake be excluded as a driver of this pattern?

RESPONSE: We agree that our original wording did not clearly convey our rationale. *S. horei* was chosen as a host because it has an evolutionary history with brood parasitic cuckoo catfish, thus making it feasibly a “challenging” host (Cohen et al. 2018, Reichard et al. 2023). No information is available on the possible natural host species of *S. granulosus*, although possible candidates would be *Cyphotilapia* spp., *Benthochromis* spp., and *Trematocara* spp. The possibility of *S. granulosus* embryos to survive in host cichlids only became apparent after combining our results and reaching preliminary conclusions, which prompted later experimental testing.

We have reworded instances of “sympatric” in relation to *S. granulosus* and cichlids to avoid implying natural co-occurrence and instead refer to “Lake Tanganyika cichlids” where appropriate. The main logic behind using *S. horei* was to test embryo survival against a challenging cichlid host, not to imply host-specific sympatry. This rationale is now clearly articulated in the revised manuscript.

Minor specific comments:

- The use of the term Pre-adaptation: This may seem like a matter of semantics, but I would encourage the authors to reflect on their use of the term pre-adaptation. While it has been widely used historically, the term can imply evolutionary foresight. The perhaps more widely accepted term is exaptation, which conveys that a trait originally evolved for one function (or without a specific function) and was later co-opted for another.

Regardless of which term is chosen, I recommend briefly defining it at first use to ensure clarity.

RESPONSE: We now use the term exaptation and introduce it specifically when first mentioned: “...that the apparent exaptations (i.e., adaptations originally evolved outside parasitism context)...”

- L- 63: In the introduction, the hypothesis: “We hypothesized that the origin of brood parasitism in the cuckoo catfish has primarily arisen from the predation of adult catfish on cichlid eggs during spawning” does feel somewhat abrupt since there is little lead-up to the idea that predation might be the key precursor.

RESPONSE: We have now reworded the statement and it starts: “Given that cuckoo catfish are potent egg predators, we hypothesised...”

The fact that the cuckoo catfish are potent egg predators is also stated earlier in the paragraph “As adult cuckoo catfish are also vigorous egg predators (15), ...”

- Figure 1: The color coding of *Synodontis* species is inconsistent across panels (e.g., in Fig. 1a *S. granulosus* is dark blue, whereas in Fig. 1b it appears purple)?

RESPONSE: We have corrected all colour biases. They resulted from variable transparency values applied in some panels where overlapping data points were visualized.

- L.93: It is mentioned that fish eggs were found in stomach contents, but I cannot locate them in the overview of dietary items shown in Fig. 1b. If they are included under a broader category, please clarify this in the text or figure legend.

RESPONSE: Fish eggs were originally included within the broader “fish” category, but we recognize that separating them in the text may have caused confusion. We have now modified Figure 1b to display fish eggs as a distinct category (brown colour) and retain this separation throughout the analysis. This clarification is also reflected in the figure legend and represented the results tables.

- Figure 1b: Cladocera are listed as prey items. To my knowledge, water fleas do not occur in Lake Tanganyika. Could this be a taxonomic mix-up, or is there a specific explanation for their presence? Clarifying this would be helpful.

RESPONSE: Upon revisiting our samples and notes, we confirmed that the items labelled as Cladocera were misidentified: one instance represents Copepoda and the other Caridea. However, we note that there is at least one report of Cladocera in Lake Tanganyika (Harding, J. P. (1957). Crustacea: Cladocera. Exploration hydrobiologique, 1946-1947) and we also observed two confirmed cases of Cladocera in the diet of *Synodontis melanostictus*, a riverine *Synodontis* occurring near the river mouths. This species was not included in our analyses and its diet may have included items from riverine rather than strictly lacustrine sources. We have corrected the figure and text to reflect these clarifications.

- L.93–94: The statement “However, the small size and spherical shape of consumed eggs clearly indicated the eggs did not belong to a mouthbrooding cichlid species.” may be a bit too general. For instance, the mouthbrooder *Perissodus microlepis* has very small, nearly spherical eggs. Perhaps clarify that they were evidently not from the major known host species rather than ruling out all mouthbrooders?

RESPONSE: We see that the text may have been misleading. We now add egg size and their large abundance but also clearly state “did not belong to any known host cichlid species.” Revised text (additions in bold): “However, the small size (<1 mm) and spherical shape of consumed eggs, **along with their high abundance (257 eggs)**, clearly indicated the eggs did not belong to **any known host cichlid** species.

- L.100–101: How was it determined whether the bones or scales belonged to cichlid or non-cichlid species?

RESPONSE: We have clarified that the scales may have also belonged to *Lates* species. Identification was based on scale morphology: all observed scales were ctenoid, and we compared them to reference collections and published descriptions of lake fish lineages (sardines, cyprinids, killifish, alestids, and catfish; Viertler et al., 2021; Jawad, 2015). To improve transparency, we have included supplementary material illustrating the fish scales (and bones, otoliths and tooth) found in the stomachs of *S. granulatus* (Suppl. Fig. 1).

RESPONSE TO REVIEWERS' COMMENTS

Reviewer #1 (Remarks to the Author):

The revised version of the manuscript addresses all of the concerns I raised in my previous review in a thorough and satisfactory manner.

Reviewer #2 (Remarks to the Author):

The authors have now thoroughly revised the manuscript and have addressed all of my previous comments. In particular, they have added phylogenetic comparative perspectives, included a negative control in the parasitism assay, and clarified all minor points raised. These additional analyses represent a valuable extension to the manuscript. Specifically, the ancestral state reconstructions now formally document the relative timing of trait evolution, which in the previous version was only visually inferred. Likewise, the inclusion of *S. polli* as a negative control has clearly strengthened the experimental design. Together, these changes have substantially improved the manuscript.

That said, I still have two minor follow-up comments, as outlined below.

1) ancestral trait reconstructions

I am not sure whether I agree with the statement that these particular analyses “suffer” from the single evolutionary origin of brood parasitism (e.g. L173–174: “Despite the interpretative limitation imposed by a single evolutionary origin of brood parasitism, significant ...”). My original concern about the effective comparative sample size of $N = 1$ applies specifically to formal tests of correlated evolution between traits and parasitism. In other words, the limitation imposed by a single origin of brood parasitism constrains analyses that would ask whether parasitic versus non-parasitic lineages differ in trait values once shared ancestry is accounted for (e.g. phylogenetic ANOVA or PGLS with brood parasitism as a predictor). Such analyses are not conducted here – instead the focus is on differences between species. While I think the manuscript is in general carefully worded not to oversell it as a test for evolutionary correlations, there are (still) instances where it is implied. For example, in the reporting of statistical results in Fig. 2 the caption states: “Asterisks indicate whether clutches of the cuckoo catfish were significantly smaller than those of all other species (LM, $P < 0.001$).” If the question is whether *S. multipunctatus* differs from other species individually, we expect pairwise comparisons not one statistical test. However, if the question is whether parasitic species differ from non-parasitic species as a group, then a comparative model accounting for phylogenetic non-independence would be required. I therefore recommend revisiting the wording throughout the manuscript to ensure that this distinction is consistently maintained and that statistical tests are described in a way that matches the questions they address.

RESPONSE: We have deleted the first part of the sentence reporting analysis of phylogenetic signal: “Despite the interpretative limitation imposed by a single evolutionary origin of brood parasitism,”, as suggested.

We have also reworked the contrast of results presented on Fig. 2 (reproductive traits). They were treated as pairwise contrast of ANOVA result and we now replace it with overall post-hoc tests (Sidak method). This does not change the outcome at all, but we have reworded the interpretation as suggested by the referee (minor changes/simplification of the main text, and figure 2 caption specification, all tracked in the marked-up version of the ms).

2) experimental parasitism

Despite the fact that *S. granulosus* performs more similarly to the cuckoo catfish than to the negative control in a naïve host, the interpretation of these results does not seem to have been fully adjusted. In light of these findings, one of my questions from the previous round remains: how can host specificity within the lake be excluded as a driver of the observed patterns?

Host–parasite interactions and adaptations go both ways. The authors argue that *S. horii* has evolved defence mechanisms through coevolution with *S. multipunctatus*. It seems equally plausible, however, that *S. multipunctatus* is specialized on its natural shallow-water hosts (shallow-water haplochromines). We do not know whether *S. multipunctatus* would perform equally well in a deep-water, non-haplochromine mouthbrooder — a “challenging” but potentially not compatible host. Basically, the reciprocal scenario relevant if *S. granulosus* were a brood parasite of deep-water hosts.

Relatedly, several arguments used to reject brood parasitism in *S. granulosus* appear to implicitly take shallow-water haplochromines as the relevant reference. For example, statements that its eggs do not visually imitate host eggs overlook the considerable variation in egg size, colour, and clutch structure among mouthbrooding cichlids. Adaptation to different hosts could plausibly result in different egg phenotypes. Similarly, arguments based on large clutch size do not fully exclude alternative scenarios, such as egg distribution across multiple hosts e.g. during lek-like spawning events.

Having said that, I want to emphasize that I am not arguing that *S. granulosus* is necessarily a brood parasite. Rather, I think the current results make it difficult to rule out host-specific parasitic strategies and suggest that traits that are expressed differently can still be relevant to brood parasitism. I therefore encourage the authors to discuss the evidence for and against parasitism in *S. granulosus* in a more balanced way (e.g. L 264-273).

RESPONSE: We have softened our arguments and mention the possibility that *S. granulosus* is specialised on deepwater cichlid hosts.

We thank both referees once more for constructive feedback and reviews.

Please note that we re-analysis of post-hoc comparisons rendered updated version of data analysis, which we resubmitted to Figshare repository as a new version.

We append **tracked version of the main ms and figure captions** (including editorial request for formatting changes and more details to provide) below.

Tracked version of **Supplementary Material** is provided as a separate file (formatting changes only, large file size due to inclusion of Figures).

The ecological and developmental foundations of brood parasitism in a catfish

Martin Reichard^{1,2,3,*}, Radim Blažek^{1,3}, Matej Poláčik¹, Tomáš Suchánek⁴, Gernot K. Englmaier¹, Kacper Pyrzanowski^{1,2}, Veronika Bartáková¹, Jakub Žák³, Lukas Koch^{1,3}, Iva Dyková³, Robert Černý⁴, Stephan Koblmüller⁵, Holger Zimmermann^{1,5,*}

odstranil: ⁵.

¹ Institute of Vertebrate Biology, Czech Academy of Sciences, Květná 8, Brno, Czech Republic

² Department of Ecology and Vertebrate Zoology, University of Lodz, Banacha 12/16, 90-237 Lodz, Poland

³ Department of Botany and Zoology, Faculty of Science, Kotlářská 2, Masaryk University, Brno, Czech Republic

⁴ Department of Zoology, Faculty of Science, Charles University in Prague, Czech Republic

⁵ Institute of Biology, University of Graz, Universitätsplatz 2, 8010 Graz, Austria

*Correspondence: reichard@ivb.cz (MR), holger.zimmermann@uni-graz.at (HZ)

odstranil:

Abstract

Interspecific brood parasitism has evolved repeatedly from parental care. However, many non-avian brood-parasitic lineages have ancestors lacking parental care, leaving the foundations of brood parasitism in these lineages enigmatic. We examine ecological, reproductive, and developmental data from the Lake Tanganyika radiation of *Synodontis* catfishes, where one species, the cuckoo catfish, exhibits brood parasitism of mouthbrooding cichlid fishes. Our comparative analyses, coupled with experimental parasitism, suggest that a combination of ancestral traits (large eggs and rapid embryo development) enabled the origin of brood parasitism. Evolutionary innovations then presumably enhanced the success of brood parasitism after it emerged. Novel traits comprise frequent production of small clutches to effectively utilize host availability, the evolution of egg mimicry to facilitate host egg adoption, and modifications to development to enhance the performance of catfish embryos in the host's buccal cavity. Interestingly, despite the distinct ecological and life history contexts of the origin of catfish brood parasitism, its evolutionary and developmental outcomes align with those of canonical avian brood parasites. This suggests that general patterns are repeated in the evolution of brood parasitism, despite disparate starting conditions.

odstranil: d

Introduction

A loudly begging cuckoo chick, tirelessly fed by a pair of tiny songbird hosts, serves as a compelling illustration of brood parasitism. Obligate brood parasites—species that usurp the parental care of other species—have independently evolved seven times in birds, with over 100 bird species classified as obligate brood parasites ¹. Theoretical and empirical insights from

research on brood parasitism have influenced our understanding of coevolutionary dynamics¹⁻⁶ and life history evolution^{7,8}, though this research has been largely confined to avian model systems where parental care is ancestral to brood parasitism⁶. However, interspecific brood parasitism is more ubiquitous in nature^{1,6}, with one-third (19 of 59) of cases found in lineages lacking parental care, with a transition to brood parasitism instead preceded by trophic ecological relationships such as saphrophagy, scavenging and parasitoidy⁹. Hence, to better understand the conditions that facilitate the evolution and persistence of brood parasitism, it is imperative to examine the origins and early stages of brood parasitism in lineages that lack parental care.

In this study, we combined ecological, reproductive, and developmental datasets across the Lake Tanganyika (LT) *Synodontis* catfish (Siluriformes, Mochokidae) radiation along with additional riverine *Synodontis* species to examine the origin and early evolution of brood parasitism in the cuckoo catfish, *Synodontis multipunctatus*. While other *Synodontis* species scatter their eggs over the substrate without parental care¹⁰, the cuckoo catfish is an obligate brood parasite of mouthbrooding LT cichlid fishes¹¹. Many LT cichlids tend their eggs and embryos in their buccal cavity (termed mouthbrooding)¹². Cuckoo catfish exploit this advanced mode of parental care by intruding on spawning cichlid pairs and spawn themselves^{13,14}. As adult cuckoo catfish are also vigorous egg predators¹⁵, the spawning female cichlid quickly responds to the risk of egg predation by hastily picking up her eggs, inadvertently collecting some eggs deposited by the intruding cuckoo catfish¹⁶. Cuckoo catfish offspring is then brooded in safety alongside the cichlid eggs in the female's mouth. The catfish eggs develop faster than those of the host and, after hatching, feed on the host's offspring^{17,18}. This system resembles the relationship between cuckoos and their hosts, with the host deceived into caring for the cuckoo's offspring and incurring a significant cost to its own reproductive success. Given that adult cuckoo catfish are potent egg predators, we hypothesized that the origin of brood parasitism in the cuckoo catfish has primarily arisen from the predation of adult catfish on cichlid eggs during spawning¹⁹. Here, we investigate the ecological, reproductive, and life history background of the LT *Synodontis* to identify traits associated with the origin and evolution of brood parasitism in *Synodontis*.

The cuckoo catfish is part of a small radiation of LT *Synodontis*. This lineage consists of two clades (Fig. 1a). The first (Clade I) comprises the cuckoo catfish (*S. multipunctatus*) and deepwater (depth >40 m) *S. granulatus*. The second (Clade II) includes *S. petricola*, *S. irsacae*, and *S. polli*, all commonly found in rocky habitats at depths from 5 m alongside the cuckoo catfish^{22,24}. The age of the LT *Synodontis* radiation has been estimated to be 2.7 (1.7–3.9) million years ago (Mya), with similar dating for further diversification within Clade I (1-2 Mya) and Clade II (0.9-1.9 Mya)²⁵.

We demonstrate that cuckoo catfish are trophic generalists rather than specialized egg predators. Their reproductive traits combine ancestral and novel characteristics that facilitate brood parasitism and consist of small batches of large, non-adhesive eggs that appear to mimic the large yellow eggs of their hosts. Developmental adaptations of the cuckoo catfish include enlarged jaws, pharyngeal teeth and associated skeletal supports, as well as accelerated mineralization of their dentition and supporting skeleton, enabling them to grasp and puncture host offspring and feed on their nutrient-rich yolk inside the cichlid buccal cavity. Many of

these traits likely evolved in other contexts but were later co-opted to enhance the brood parasitic strategy of the cuckoo catfish.

Results

Brood parasitism in the cuckoo catfish is not associated with trophic specialization

Brood parasitism may evolve from a trophic relationship between parasitic and host species, including the exploitation of host eggs⁹. Furthermore, many evolutionary radiations originated from dietary specialization^{12,26,27}. Thus, we examined gastrointestinal tracts and employed stable isotopic analysis to investigate whether the cuckoo catfish occupies a specific trophic niche within LT *Synodontis*. Considering that host egg predation is a feasible stepping stone to interspecific brood parasitism^{9,19}, we hypothesized that the cuckoo catfish specializes to feed on nutritionally rich cichlid eggs.

First, we analyzed the diets of all five LT *Synodontis* species collected in the lake. Dissecting the gastrointestinal tracts of 122 specimens and calculating Bray-Curtis dissimilarities (BCDs) based on volumetric estimates of their diet revealed that the primary differences among LT *Synodontis* species were associated with the relative contribution of sponges, benthic insect larvae, fish tissue, and algae (Fig. 1b, Likelihood Ratio tests: Table S1, SIMPER analysis: Table S2). The diet of the cuckoo catfish quantitatively differed from those of all other LT *Synodontis* species (multivariate GLM, $P < 0.001$; Table S2), and uniquely amongst LT *Synodontis*, the gut of one (of 30 examined) cuckoo catfish indeed contained fish eggs. However, the small size (<1 mm), spherical shape and high abundance ($n = 257$) of consumed eggs clearly indicated the eggs did not belong to any cichlid species confirmed to be host of the cuckoo catfish²⁸. The cuckoo catfish is obviously not a specialized egg predator. Instead, its diet is very generalized; its gastrointestinal tracts contained a mix of dietary components, featuring a high proportion of benthic aquatic invertebrates from soft sediments (Fig. 1b). This is despite that three other LT *Synodontis* species are clear dietary specialists (Fig. 1b); the variation in their within-species BCD values is negligible compared to between-species BCDs (<0.32, Fig. 1c, Table S4). The diet of *S. granulatus* consists solely of fish tissue (bones and ctenoid scales of either cichlids or *Lates*, Fig. S1)^{29,30}, typical of predators and scavengers. In contrast, *S. polli* is an herbivore, feeding on algae and plant detritus. The third specialist (*S. irsacae*) fed primarily on sponges (Porifera) (Fig. 1b, c). Like the cuckoo catfish, *S. petricola* is identified as a dietary generalist (Fig. 1b, c; Table S4).

The examination of gastrointestinal tracts offers only a short time window for food consumption. To compare time-integrated trophic ecology, we analyzed signatures from stable isotopic (SI) ratios of $\delta^{13}\text{C}$ and $\delta^{15}\text{N}$ in muscle tissues from 198 catfish. The results confirmed substantial trophic niche separation among LT *Synodontis* (Fig. 1d, Table S5). All species differed in their trophic position, except for the similar SI signatures observed in the cuckoo catfish and *S. petricola* (PERMANOVA: Table S6), two trophic generalists. The trophic level of *S. granulatus* was higher than that of a specialist predatory cichlid (Fig. 1e), suggesting that *S. granulatus* is a fish scavenger rather than a predator³¹. The herbivorous *S. polli* encompassed the combined niches of two herbivorous cichlids¹² and sponge-eating *S. irsacae* occupied an extreme dietary niche with no known LT cichlid equivalent (Fig. 1e). The SI trophic niche of the cuckoo catfish was broadest of all LT *Synodontis* species (Fig. 1d). When compared to the trophic niche space of the cichlid radiation in Lake Tanganyika, which is strongly underpinned

odstranil: Extended Data

odstranil: Supplementary

odstranil: 1

odstranil: Extended Data

odstranil: 2

odstranil: Extended Data

odstranil: 3

odstranil: Supplementary

odstranil: Extended Data

odstranil: 3

odstranil: Extended Data

odstranil: 4

odstranil: Extended Data

odstranil: 5

by trophic ecology¹², the cuckoo catfish largely overlapped with the SI space of several cichlid species (Fig. 1e). This further discounts the hypothesis of its trophic specialization on fish eggs, which would be associated with a higher $\delta^{15}\text{N}$ signature³².

Both diet and SI analyses consistently demonstrate a trophic component in the LT *Synodontis* radiation. However, the generalized diet of the cuckoo catfish and the rare occurrence of fish eggs in its diet undermine our hypothesis that the cuckoo catfish is a specialized egg predator. While the detection window for eggs in the gastrointestinal tract is short³³ and may lead to underestimation, the gastrointestinal tracts of West African *Synodontis* species frequently contain fish eggs³⁴. Therefore, strict trophic specialization is not a key driver in the origin of brood parasitism in this lineage.

The reproductive traits of cuckoo catfish combine ancestral and novel characteristics that facilitate brood parasitism

One of the biggest challenges for brood parasites is to synchronize their reproduction with that of their hosts and to induce them to accept parasitic eggs. We predicted that the cuckoo catfish produces multiple small clutches to match the continuous breeding season of their hosts³⁵ and thus be ready to capitalize on forthcoming opportunities for parasitism. We also predicted that cuckoo catfish have evolved egg characteristics that facilitate host acceptance through mimicry in terms of egg size and coloration. The usual *Synodontis* spawning is seasonal and involves egg scattering, during which the eggs may adhere to a substrate¹⁰. Since egg adherence may hinder collection by host cichlids and might compromise developmental performance in the host's buccal cavity, we predicted decreased egg adhesiveness of cuckoo catfish eggs. We used hormonally-induced reproduction in five LT *Synodontis* and three additional riverine *Synodontis* species to collect data on standardized clutch and egg characteristics. We further obtained histological data on the gonads of four LT *Synodontis* species from individuals collected in the wild at the start and end of the rainy season (n = 239).

Notably, the clutch size of the cuckoo catfish (mean 28, range 12-45 eggs) was an order of magnitude smaller than that of all other *Synodontis* species (means of 208-767 eggs) (LM, $P < 0.001$, Fig. 2a, Table S7). The size of individual cuckoo catfish eggs (2.7 mm) was approximately double that of other LT (1.3-1.8 mm) and riverine (1.3-1.5 mm) *Synodontis* (LM, $P < 0.001$), except for its sister species, *S. granulatus*, which produced slightly larger eggs (2.8 mm) (Fig. 2b, Table S8). *Synodontis* egg size was not related to body size, as the cuckoo catfish (female total length 83 ± 1.9 mm) and *S. petricola* (87 ± 1.9 mm) were smaller than *S. irsacae* (99 ± 3.0 mm), *S. polli* (120 ± 1.7 mm) (Fig. 2c, d) and riverine *Synodontis*¹⁰, as well as *S. granulatus* (131 mm), the largest LT *Synodontis* species. The color of cuckoo catfish eggs was distinctly shifted towards a yellow hue compared to the more transparent eggs of all other LT *Synodontis* and the opaque brownish eggs of riverine *Synodontis* (LM, yellow-blue CIE_B axis: $P < 0.001$; red-green CIE_A axis: $P \leq 0.03$; Fig. 2e, Table S9). This modification of the cuckoo catfish egg color and size appears to imitate notably larger, yellow eggs of host cichlids^{35,37}. Finally, while the eggs of three LT *Synodontis* were highly adhesive (50-95% remained adhered to a vertical surface), the cuckoo catfish and *S. granulatus* laid eggs with considerably lower adhesiveness (20-25% adhered; ordered-beta GLM, $P < 0.001$). Low adhesiveness was not exclusive to the cuckoo catfish and *S. granulatus*, as it was also recorded in some riverine *Synodontis* (Fig. 2f, Table S10). We also conducted ancestral trait reconstructions and calculated

odstranil: for all pairwise contrasts with the cuckoo catfish

odstranil: Supplementary

odstranil: XX

odstranil: all pairwise contrasts

odstranil: Supplementary

odstranil: XX).

odstranil: and

odstranil: , *S. granulatus* (131 mm)

odstranil: Extended Data

odstranil: 6

odstranil: Supplementary

odstranil: XX

evolutionary signal metrics (Pagel's λ and Blomberg's K) for reproductive traits. Significant phylogenetic signal was detected in egg coloration along the red–green axis, but not the yellow–blue axis, and moderate phylogenetic signal was observed for clutch size (Fig. S2).

Reproductive seasonality, based on the histological examination of the gonads of fish collected in the lake, was observed in a single species, *S. irsacae*, with 0% of females and 40% of males exhibiting ripe gonads in November and 93% and 100% in March (n=56; χ^2 tests: $P < 0.001$ and $P = 0.003$). In contrast, other LT *Synodontis* had 67–90% of fish with ripe gonads across sexes and species in November, and 94–100% in March (n=182 across three species; Table S11). The relative female investment in reproduction, calculated as ovary mass relative to body size in fish with ripe gonads, was highest in the seasonally spawning *S. irsacae* (GLM, $P < 0.05$), but comparable among other LT *Synodontis*, including the cuckoo catfish (Fig. 2c). Relative testis mass did not differ among species (GLM, $P = 0.185$; Fig. 2d).

The analysis of reproductive traits thus revealed that cuckoo catfish produce small batches of large, non-adhesive eggs that mimic the large yellow eggs of their hosts³⁵ which likely facilitates egg adoption by the hosts. Further, the large cuckoo catfish eggs contain large yolk (nutrients) reserves to support rapid early development³⁸. The regular production of small egg batches uniquely enables the cuckoo catfish to spawn every few days¹⁴, while other traits conducive to brood parasitism (large egg size, low egg adhesiveness, and lack of reproductive seasonality) are not specific adaptations to a parasitic strategy but rather preceded its origin.

The early development of the cuckoo catfish has evolved to exploit the host clutch

In the host buccal cavity, cuckoo catfish hatch early and, upon consumption of their own yolk sac, prey on the host clutch¹⁷. We predicted that cuckoo catfish embryos possess adaptations for predation on host offspring inside the cichlid buccal cavity, specifically in terms of the accelerated development of their dentition and jaw apparatus¹⁷. We hormonally induced the reproduction of all five LT and two riverine *Synodontis* species and produced a standardized fine-scale ontogenetic series following *in vitro* fertilization. We then compared early growth and development, with particular emphasis on the oral and pharyngeal dentition.

All *Synodontis* species hatched within the first 2 days post-fertilization (dpf) and consumed their embryonic yolk sac at 6 dpf (Fig. S3). The cuckoo catfish and *S. granulosis* exhibited higher post-hatching growth rates than all other species (Fig. 4a). The offspring of these two species were nearly twice as large (LM, $P < 0.001$, Table S12) and considerably more developmentally advanced at the age of 10 dpf (Fig. 4b). However, only the cuckoo catfish possessed a well-mineralized dentition at 6 dpf (Fig. 4c) when they started to consume host offspring^{17,18}. The cuckoo catfish also initially possessed larger jaws (Fig. 4c), pharyngeal teeth, and associated skeletal support (Fig. 4d, Fig. S4), as well as accelerated mineralization of its dentition and supportive skeleton (Table S13) in comparison to other LT *Synodontis*, including *S. granulosis*. *Synodontis granulosis* initially had a slightly larger pharyngeal dentition than other species (Fig. S4, Table S13), but the pace of its development and mineralization did not differ from the other species.

Comparison with the rate of dental development in other teleost fishes clarified that the development of the cuckoo catfish's dentition and supportive skeleton is strongly accelerated^{39,40}. We propose that the cuckoo catfish has evolved a uniquely rapid post-hatching development, which enables its embryos to grasp and puncture host offspring to feed on their

odstranil: Despite the interpretative limitation imposed by a single evolutionary origin of brood parasitism,

odstranil: s

odstranil: Extended Data

odstranil: 1

odstranil: Extended Data

odstranil: 7

odstranil: all pairwise contrasts $P > 0.60$;

odstranil: Supplementary

odstranil: 2

odstranil: Supplementary

odstranil: Extended Data

odstranil: 2

odstranil: Extended Data

odstranil: 8

odstranil: Extended Data

odstranil: 2

odstranil: Extended Data

odstranil: 8

nutrient-rich yolk. These adaptive modifications of the cuckoo catfish derive from pre-existing traits, such as larger eggs and a more rapid post-hatching growth, which it shares with its sister species *S. granulosus*. However, *S. granulosus* do not possess dental and skeletal adaptations to feed on host embryos. These observations corroborate our hypothesis that the embryo development of the cuckoo catfish is under strong selection to enhance its ability to consume host clutches, resulting in the rapid development of embryonic dentition.

Potential for brood parasitism in *S. granulosus*

Considering that large egg size and rapid embryo growth are shared between the cuckoo catfish and its sister species *S. granulosus*, it might be speculated that *S. granulosus* represents an unrecognized brood parasite. Unlike all other LT *Synodontis* species, which are commonly bred in captivity and documented as egg scatterers^{10,23}, there is a lack of data on the reproduction of deepwater *S. granulosus*. Hence, we examined whether *S. granulosus* eggs and embryos can survive, develop, and feed in the buccal cavity of a host cichlid and repeated the same experiment with *S. polli* as a non-parasitic species control.

We experimentally inoculated cichlids¹³ with *S. granulosus* eggs and tested their survival over the incubation period of 4, 7, 12 and 14 days, using the cuckoo catfish eggs inoculated into other mouthbrooding females as controls. First, we used a Lake Victoria cichlid (*Haplochromis* sp. CH44), a non-sympatric cichlid species which has not evolved egg rejection of the cuckoo catfish¹³. We found that the survival rate of *S. granulosus* in mouthbrooding *Haplochromis* was high (75% of 56 eggs inoculated across 15 mouthbrooding host females and all incubation periods), and consistent with survival of the cuckoo catfish embryos (73.2% of 56 eggs across 14 mouthbrooding hosts; Binomial GLMM, $P = 0.873$) (Fig. 4a, b). The ability to survive incubation in cichlid mouth is not a universal feature of LT *Synodontis*; no embryos of non-parasitic *S. polli* were recovered from *Haplochromis* hosts (20 eggs across 5 mouthbrooding hosts) after 7 days of incubation, whereas 93% (26 of 28 inoculated eggs across 7 mouthbrooding hosts) cuckoo catfish control embryos survived that period (Fig. 4c). Cuckoo catfish embryos grasp and pierce host embryos to suck their yolk sacs (Fig. 4d, e) from day 4 post-hatching and no cichlid embryos were found in experimentally parasitized females ($n = 13$) at age of 12-14 days. Despite having a comparable body size at that age (Fig. 3a, b), the dentition of *S. granulosus* embryos is relatively undeveloped (Fig. 3c, d), and the alimentary tract of *S. granulosus* offspring recovered from host buccal cavity contained only traces of material resembling yolk (Fig. 4f) while host offspring also disappeared ($n = 9$ host clutches).

When *S. granulosus* eggs were inoculated into the buccal cavity of a natural cuckoo catfish host (*Shuja horei*)²⁸ (which can effectively decrease the load of cuckoo catfish eggs⁴¹), none of the 18 eggs (across 6 mouthbrooding *S. horei*) survived 4 or 7 days of incubation, compared to a 30% survival rate of cuckoo catfish eggs (6 of 20 eggs across 4 of 6 mouthbrooding *S. horei*) (Fig. 4g). This finding suggests that the apparent exaptations (i.e., adaptations originally evolved outside parasitism context) to brood parasitism in *S. granulosus* are sufficient to survive 14 days of incubation (i.e. 8 days after yolk sac consumption) in an evolutionarily naive host, although their capacity to extract resources from the host clutch was inefficient. However, these putative exaptations to brood parasitism failed to support survival in the more challenging environment of LT cichlid which is a common host of the cuckoo catfish²⁸ and has evolved defenses against parasitism¹⁷.

If *S. granulosus* is indeed an unrecognized brood parasitic species parasitizing deep-water cichlids, it possesses many traits of the generalized *Synodontis* life history. First, its very large clutches exceed the capacity of host buccal cavity by more than an order of magnitude. Second, the eggs do not visually imitate host eggs. Third, their ovulation frequency is not adapted to the opportunistic exploitation of unpredictable opportunities to parasitism. Finally, its embryonic dentition is not adapted to efficient extraction of resources from host embryo, as seen in the cuckoo catfish. Nevertheless, by using hosts from shallow-water habitats in our experiment, we cannot rule out the possibility that *S. granulosus* has specialized to parasitise deep-water cichlids. Irrespectively of breeding strategy of *S. granulosus*, its reproductive and developmental characteristics appear to represent ancestral conditions within this small clade and perhaps exemplify the transient state, with many traits which have evolved in a different context, and from which brood parasitism in cuckoo catfish might have evolved.

Discussion

Brood parasitism has evolved multiple times in species whose ancestors exhibited no parental care. It has been hypothesized that trophic exploitation of the potential host brood (e.g., predation, parasitoids) initiates the origin of brood parasitism⁹, but it remains unclear whether the subsequent host-parasite coevolutionary arms race converges on traits analogous to those of canonical brood parasites⁸. With the limitation of testing single evolutionary origin without a replication⁴², we investigated this hypothesis in the cuckoo catfish, a brood parasite of mouthbrooding cichlids. Although cuckoo catfish often consume host eggs when they intrude during host spawning¹⁶, we revealed that they are not specialized egg predators. Occasional interactions with cichlid hosts, when feeding on their eggs, may have been sufficient to trigger the origin of obligate brood parasitism, perhaps facilitated by the exploitation of the strong maternal instinct of mouthbrooding cichlids⁴³. This highlights the importance of pre-existing traits in the host, along with those in the parasite, at the origin of brood parasitism.

The evolutionary innovations of the cuckoo catfish were formed by selection on already existing traits. These traits (large eggs and rapid embryo growth) may have evolved in a different context but were later co-opted to support innovations which enhanced the success of brood parasitism after it emerged. The cuckoo catfish innovations parallel the traits exemplified by virulent brood parasites among birds and consist of frequent production of small clutches, which allow for effective utilization of host availability^{44,45}, the evolution of egg mimicry to facilitate egg adoption^{5,46}, and developmental traits that enhance the performance of catfish embryos in the host's buccal cavity^{4,47} by enabling them to harm host offspring and exploit host resources^{1,8,48}. Overall, our study demonstrated that, despite different ecological and life history foundations of catfish brood parasitism, the developmental consequences of the brood parasitic strategy in the cuckoo catfish mirror the traits central to the coevolution of avian brood parasites with their hosts.

odstranil: Conclusions

Methods

All research methods and protocols adhered to national and institutional animal care and use guidelines, administered under permit No. CZ62760203 and approved by ethical boards of the Institute of Vertebrate Biology and the Czech Academy of Sciences (approval No. 32-2019). Fieldwork and sample export have been approved by research permits (Zambia: K-4335/18 KA/K.48/18 and individual study permits of MR, HZ, RB, MP, VB and SK), Tanzania (TAFIRI/HQ/RES.CLEARANCE/82, Costech: 2022-603-NA-2022-228). Import of samples to EU has been approved by Czech State Veterinary Administration (Zambia: SVS/2024/064505-G, Tanzania: SVS/2024/064505-G).

Adult fish collection

Adult *Synodontis* and cichlids were collected during nine field expeditions to Zambia and Tanzania between 2019 and 2024. Fish were collected during Scuba dives using hand nets, stop nets, and baited minnow traps and brought to the surface for further processing. Additional samples (of deepwater *S. granulatus*) were obtained from deep-water gill nets deployed by local fishermen.

Fish were typically processed within the first two hours after collection. They were euthanized with an overdose of clove oil, and a piece of fin was stored in 96% ethanol for DNA barcoding and further genetic analyses. The fish were measured, weighed, labelled, fixed in 6% formaldehyde, and subsequently transferred to 70% ethanol for long-term storage. A piece of muscle from the fish flank (dorsolateral side) was taken and stored in ethanol for the analysis of stable isotope ratios. Vouchers are stored at the Natural History Museum in Vienna, and collections are held at the Institute of Vertebrate Biology, Czech Academy of Sciences⁴⁹. Morphological identifications were confirmed by DNA barcoding using the cytochrome c oxidase I gene²².

Phylogeny

A detailed ddRADseq-based phylogeny with broad geographic representation has been published elsewhere²². To visualize the phylogenetic background of our study species, we selected one individual for each study LT *Synodontis* species and one riverine outgroup (*Synodontis nigrita*). The tree was inferred using a maximum likelihood approach in IQ-TREE v.2.0.7 and a concatenated alignment of 6536 ddRADseq loci. We used the same nucleotide substitution model (TVM+F+R2) as in²² and 5000 ultrafast bootstrap replicates.

Dietary data and analyses

A diet analysis was conducted on samples collected from the southern tip of the lake in Zambia, specifically at Mpulungu (Chituta Bay, Kalambo Lodge, Mutondwe Island, Mpulungu) and Ndole Bay (Chimba Rocks, Mpende Fisheries, Katete, and Cape Kachese), where all examined fish species co-occur locally. All primary sample data are georeferenced⁴⁹. For the diet analysis, we included only fish processed within two hours of their collection. Individuals with unidentified digested matter (i.e. with fish bait used in the traps) in their gastrointestinal tract were excluded from further analyses.

In the laboratory, all specimens were measured for total length (TL) to the nearest 1 mm and weighed (W) to the nearest 0.01 g. Gut contents were weighed to the nearest 0.001 g and stored in glycerin. Food items were subsequently identified to the lowest practical taxon (i.e., to order, family, genus, or species) under a stereomicroscope (Nikon SMZ1000, Japan) and counted, or their counts estimated in cases of high abundance of particular prey items. The total number and biomass (to the nearest 0.1 mg) of each prey type were recorded for every fish specimen. Prey items were combined by taxon and quantified by frequency of occurrence and percentage of biomass. Proportions of diet were calculated from the contents of the gastrointestinal tracts, excluding the category ‘animal remains’ which were decayed beyond a state to identify it. Chironomidae, Ephemeroptera, Trichoptera, Coleoptera, and Heteroptera were pooled as ‘benthic insects’. All other diet categories were kept separate, as we could not unambiguously distinguish their ecological origins (i.e., benthic vs. planktonic, surface-bounded vs. free roaming). We analyzed 122 fish (30 *S. multipunctatus*, 3 *S. granulosus*, 41 *S. irsacae*, 9 *S. polli*, and 39 *S. petricola*).

To test species differences in diet composition, we used Multiple Response Permutation Procedure (MRPP). We calculated the average within- and between-species differences (Bray-Curtis dissimilarities, BCDs) in diet compositions (meandist; package ‘vegan’) ⁵⁰. Dietary specialists are predicted to have small average within-species BCDs, whereas the average within-species BCDs of dietary generalists is the same as or higher than the average between-species BCDs. We then examined the overall species effect on the diet using a Multivariate Generalized Linear Model (multivariate GLM) with a negative binomial error distribution (manyglm; package ‘mvabund’) ⁵¹. This approach addressed the challenge of interpreting statistical significance in distance-based multivariate comparisons, which is complicated by the confounding effect of abundance on variance in the data. Subsequently, an ANOVA was conducted to determine the importance of specific food categories in the separation across *Synodontis* species using the Likelihood Ratio statistic (LRT) (Table S1). Only four food categories (algae, sponges, zoobenthic insects, Copepoda, and fish scales) had a significant effect on the model fit (Table S1). We then employed SIMPER analysis to assess the importance of individual food categories in the pairwise contrasts among *Synodontis* species. Food categories contributing cumulatively to >80% dissimilarity within each species pair are listed in Table S2. Finally, a multivariate PCoA (Principal Coordinates Analysis) was applied to represent the BCD values in Euclidean space. Since the variables ‘locality’, ‘sex’, and ‘body size’ did not significantly contribute to the multivariate GLM, only the variable ‘species’ was included in the final analysis.

Stable isotope data and analyses

To estimate the extent of trophic niche separation among LT *Synodontis*, we measured the stable carbon ($\delta^{13}\text{C}$) and nitrogen ($\delta^{15}\text{N}$) isotope composition of 198 *Synodontis* specimens (5 species), 62 cichlids (8 species), and 34 baseline items from various trophic levels. A piece of fish muscle (or entire individuals for the baseline items) was initially stored in ethanol and dried in the laboratory. To account for a shift in the values of $\delta^{13}\text{C}$ and $\delta^{15}\text{N}$ caused by preservation in ethanol ⁵², we collected a subsample of each species as dried samples paired with the ethanol-based sample. The reported stable isotope (SI) values were corrected using the average shift in SI values for each species separately. Samples (2 mg of dry mass) were analyzed by Iso-

odstranil: Extended Data

odstranil: Extended Data

odstranil: Supplementary

odstranil: 1

Analytical Ltd. (Crewe, UK) using the Europa Scientific 20-20 IRMS elemental analyzer calibrated against international standards. The $\delta^{13}\text{C}$ values were corrected for lipid content following established protocols⁵³. Isotope values of baseline items were corrected for trophic enrichment according to specified methods⁵⁴ and organized into informative categories (i.e., macrophyta: a combination of 4 plant species). Full details are provided in the source data⁴⁹.

The $\delta^{13}\text{C}$ and $\delta^{15}\text{N}$ values were represented as biplots. Baseline and *Synodontis* values were overlaid with stable isotope signatures of cichlids (8 species with well-defined trophic positions¹², encompassing a large isospace). This allowed us to compare the degree of overlap in trophic positions of LT *Synodontis* with the locally coexisting cichlid trophic guilds. The $\delta^{15}\text{N}$ values indicate sample positions in the food chain; $\delta^{15}\text{N}$ enrichment increases with trophic levels. The $\delta^{13}\text{C}$ values help to indicate primary producers at the start of the food chain and differentiate between littoral and pelagic environments; higher values suggest shallow (littoral) sources, while lower values indicate deeper (pelagic) habitats^{31,55}.

We employed permutation-based analysis of variance (PERMANOVA) to examine differences in SI signatures among *Synodontis* species using the function 'pairwise.adonis' (R package pairwiseAdonis)⁵⁶, which acts as a wrapper for the 'adonis' function from the R package 'vegan', with 10,000 permutations. We applied the Benjamini-Hochberg procedure to account for multiple testing.

Reproductive traits

Gonads were dissected from formalin-fixed specimens (127 females, 114 males) of *Synodontis* collected at the beginning (November 2021) and final part (March 2022) of the rainy season in the Zambian region of the lake (Mpulungu area). The fish were dissected in the laboratory, the gonads were separated, weighed to the nearest 0.01 g, immersed in 70% ethanol and processed by Histoplast (Sigma). For each gonad, 60–80 semi-serial sections were examined using hematoxylin and eosin (H&E) staining under light microscopy to detect the presence of mature oocytes and spermatozoa⁵⁷. We identified a season-specific proportion of sexually mature individuals, namely fish with mature oocytes (indicated by the presence of at least some gametes at the two latest stages of vitellogenesis or spermiogenesis)^{57,58}. We compared the proportion of ripe gonads between the beginning and end of the rainy season for each species and sex separately, using chi-squared tests.

Investment in gonads was modeled as gonad mass (mg) relative to body size (TL, mm) for females and males separately. Gonad mass was log-transformed before analysis and modeled (Linear Model) as a function of body size (TL), species, and their interaction. When the interaction between TL and species was not significant (i.e., the slope of the relationship between gonad mass and TL was not species-specific), a simplified model (without interaction) was employed, and its fit was compared with the more complex model using ANOVA. Tukey tests were used for pairwise differences when species significantly differed in their relative investment in reproduction. Two female *S. irsacae* were excluded from the statistical analysis as major outliers (ovaries of 8.03 and 8.58 g compared to <2.7 g for all other females). Including these two females in the dataset would add stronger support to the outcome of larger ovaries in *S. irsacae*. The TL measured on this set of fish was used to compare body size across *Synodontis* species.

A comparison of egg traits (size, adhesiveness, coloration) among five LT species and three riverine species of *Synodontis* was conducted on eggs obtained following hormonal stimulation of fish (Ovaprim, a salmon gonadotropin-releasing hormone analog, Syndel Laboratories Ltd., Canada)⁵⁹. Females, under clove oil anesthesia, were injected with 0.5 µL/g of Ovaprim, and ripe eggs were gently stripped out after 12 to 24 hours onto glass Petri dishes. After 3 to 5 minutes, the dishes containing the eggs were carefully transferred into a larger glass tank filled with water and positioned vertically (at 90°). Egg adhesion was determined as the proportion of eggs remaining on the glass surface after 5 seconds compared to the total number of eggs initially placed on the dish. Eggs that did not adhere to the glass surface fell away within this 5-second period. Egg adhesion was modelled as the proportion of eggs adhered to the Petri dish in a vertical position relative to the total number of eggs on the dish, using a Generalized Linear Model (GLM) with ordered beta distribution and a logit link. The egg adhesion of the cuckoo catfish was set as the reference value, and contrasts with the egg adhesion of other species were tested.

odstrani: n

To compare egg coloration, a subsample of eggs from each female was placed on a separate glass Petri dish. Standardized pictures (RAW format) were taken over a standard grey card (18% standard reflectance, KELDAN grey card including a scale tape) from each subsample of eggs immediately after the test for egg adhesiveness. Raw pictures were adjusted to the grey background in the Mica ToolBox (ImageJ plugin,⁶⁰ and converted into a linear multispectral image. The linear images were transformed into cone-catch images (representing the human visible range in daylight) by using a custom-prepared cone-catch model for the digital camera used (Canon 7D). This was carried out by taking a RAW image of the KELDAN color checker under natural daylight, preparing a linear multispectral image of the color checker, and following the standard protocol and the built-in algorithm of the Mica ToolBox to generate a cone mapping model (using the CIE 1931 400–700 nm, D65 400–700 nm configuration). The cone-catch images were then further transformed into CIELab color space within the Mica ToolBox. Seven to ten eggs were measured per individual clutch (except for only three eggs in a *S. macrostigma* clutch). The average values for each clutch were used in further analysis. Egg coloration was quantified by CIE_A and CIE_B axes. The CIE_A represents the red-green continuum (positive values: red, negative values: green, zero: grey). The CIE_B represents the yellow-blue continuum (positive values: yellow, negative values: blue, zero: grey). A Gaussian Linear Model was used to compare the CIE_B and CIE_A among species, with the cuckoo catfish set as the reference species. The contrasts between the cuckoo catfish and the other species are reported.

To analyze egg size, the images taken for color comparison (which included a scale) were used, and the egg diameters were measured using ImageJ 1.53t. A Linear Mixed Model (LMM) with Gaussian error distribution compared egg sizes across species. Up to 10 eggs were measured from the same clutch, with clutch ID modelled as a random intercept to account for the lack of independence. The cuckoo catfish served as the reference species.

odstrani: , and contrasts to the other species are reported

Clutch size was estimated from the same set of clutches as other traits (i.e., following hormonal stimulation). The eggs were counted on Petri dishes. A linear model using log-transformed data was employed to calculate interspecific contrasts. We also compared the clutch sizes obtained during natural egg maturation and following hormonal stimulation in the cuckoo catfish (where comparison was feasible), finding no difference (Linear Model, P =

odstrani: contrasts between the cuckoo catfish and the other species

0.728, N = 9 and 10 clutches for hormonally induced and natural clutches, respectively). All statistical analyses were conducted using *glmmTMB* ⁶¹. We tested model dispersion, zero inflation, and model misspecification using the *DHARMA* package ⁶². Post-hoc pairwise comparisons were calculated using *emmeans* ⁶³ with Benjamini-Hochberg procedure for multiple testing.

We then used the *phytools* package ⁶⁴ to reconstruct ancestral character states and the *geiger* package ⁶⁵ to calculate evolutionary signal metrics. Blomberg's K measures the relative strength of phylogenetic signal, estimating how closely related species resemble each other compared with expectations under Brownian Motion (BM) evolution. We used Pagel's λ to estimate the extent to which the phylogeny explains trait covariance. *Synodontis nigriventris* and *S. macrostigma* were used as outgroups.

odstranil: ⁶⁴

Developmental traits

To compare the growth and development of LT *Synodontis*, we conducted *in vitro* fertilization in the laboratory to raise the offspring under standardized conditions. Wild-caught adult fish were treated with Ovaprim, according to ⁵⁹, with species-specific modifications in timing. The eggs and sperm were extracted from the adult fish and mixed for a minimum of 10 min. The fertilized eggs were placed in a water recirculation system and maintained at 27 ± 0.5 °C. One day post-fertilization, all non-developing (i.e., unfertilized) eggs were removed. Embryo samples were collected at 24-h intervals until the age of 10 days post fertilization and stored in 4% PFA at 4 °C.

The body size of fish was measured from the snout to the end of the caudal fin (Total Length, TL) to the nearest 0.1 mm. Growth rate was modelled using a Gaussian Linear Model on log-transformed TL, with species, age, and their interaction as fixed factors in the *glmmTMB* package. Species-specific contrasts were also analyzed at 10 dpf.

To assess the development of dentition, PFA-stored embryos of all LT and two outgroup *Synodontis* species were washed in PBS for 10 min at room temperature (RT) and then bleached in a solution containing 1% KOH and 3% hydrogen peroxide (1:1) under light. Subsequently, the specimens were washed in distilled water for 30 min at RT and placed in a saturated sodium tetraborate (Sigma-Aldrich: cat. No. 71996) solution overnight. Further staining for mineralized tissues was carried out in Alizarin Red S (Thermo Scientific Chemicals, cat. No. 400481000) solution in 1% KOH overnight, followed by washing in distilled water for 30 min. Embryos were then transferred to 1% trypsin (Sigma-Aldrich, cat. No. T4799) in 2% sodium tetraborate solution for 1-3 days (depending on the embryo size). After that, specimens were moved into glycerol through a series of glycerol in 1% KOH solution (25%, 50%, 75%) with each step performed overnight. Once in glycerol, the specimens were squeezed and coverslipped. Photographs were taken using an Olympus BX51 microscope under fluorescent light with an Olympus DP74 camera, employing the cellSens software. Final images were composed in Photoshop 2023 from photos of varying focus distances; contrast and colors were adjusted.

For skeletal development, staining followed the protocol by ⁶⁶. Photographs were taken using an Olympus SZX12 stereoscopic microscope equipped with an Olympus u-tv0.5xc-3 camera and QuickPhoto Micro (ver. 2.3) software, during which Z-stacking was also performed. Final images were edited in Photoshop to smooth the background and minimize distraction.

odstranil: ⁶⁵

Potential for *S. granulosis* parasitism

Based on the reproductive and developmental traits of *S. granulosis*, we experimentally assessed whether the existing set of traits enables its offspring to survive in the buccal cavity of a mouthbrooding cichlid. Six pairs of wild-caught adult *S. granulosis* were hormonally stimulated (Ovaprim, as detailed above) for in vitro fertilization to obtain viable embryos. The fertilized eggs were then placed in tumblers with recirculating incubation water²⁸. Water for incubation was prepared using reverse osmosis water with 11 g/L Cichlid Lake Salts (Seachem Laboratories Inc., USA). The pH was adjusted to 8.5–9 using Tanganyika Buffer (Seachem Laboratories Inc., USA), and the temperature was set to 27 (\pm 0.5) °C. The eggs of *S. multipunctatus* (used as parasitic controls) were obtained from 10 pairs and treated in the same manner as the eggs of *S. granulosis*. The eggs of non-parasitic control *S. polli* were obtained from 9 pairs of hormonally stimulated parents. As the experiments with *S. granulosis* and *S. polli* were conducted at different time, a separate control using *S. multipunctatus* eggs (obtained from 10 parental pairs) was completed during *S. polli* experiment. After 24 hours of incubation, the experimental eggs were examined under a binocular microscope to confirm their proper morphological development. The same water quality was used for incubation and subsequent experiments.

To maximize the simultaneous availability of freshly spawned mouthbrooding females, we separated the sexes of both host species (*S. granulosis* experiment: 54 females and 19 males of *Haplochromis* sp. CH44; 30 females and 15 males of *S. horei* for two weeks; *S. polli* experiment: 69 females and 16 males of *Haplochromis* sp. CH44 for three weeks). Subsequently, we housed females and males together for three days and used any freshly mouthbrooding cichlid female from the two species for inoculation. *Haplochromis* sp. CH44 is a species from Lake Victoria, known to be susceptible to experimental brood parasitism by *S. multipunctatus* eggs and embryos¹³ and producing relatively small eggs (2.5 mm). *Shuja horei* is a natural host of *S. multipunctatus*, which likely possesses defense mechanisms⁴¹ and produces larger eggs (4.7 mm). We used two or four developing *Synodontis* eggs at 24 hours post-fertilization to artificially inoculate a mouthbrooding female. The eggs were collected into a Pasteur pipette, which was then gently inserted into the mouths of the brooding cichlid female and released¹³.

Inoculated host females were transferred to 60 L housing tanks, which were equipped with a perforated false bottom (mesh size 0.5 cm), an air-driven sponge filter, and a clay pot serving as a shelter. The tanks were dimmed to minimize external disturbances. The females were allowed to incubate their clutches for 4 days (*S. horei* host: 3 clutches per *Synodontis* species; *Haplochromis* host: 2 clutches of *S. granulosis*) and 7 days (*S. horei* host: 3 clutches per *Synodontis* species; *Haplochromis* host in *S. granulosis* experiment: 4 clutches of *S. granulosis* and 7 clutches of *S. multipunctatus*; *Haplochromis* host in *S. polli* experiment: 5 clutches of *S. polli* and 7 clutches of *S. multipunctatus*). Because *S. polli* embryos had zero survival in the benign *Haplochromis* hosts, we did not inoculate their eggs into more challenging host (Tanganyikan *S. horei*). In contrast, the high survival rate of *S. granulosis* in *Haplochromis* hosts led us to extend the inoculation of 3 clutches to 12 days to test whether the offspring of *S. granulosis* can survive in the host's buccal cavity after all resources from their own yolk sac were depleted (which occurs at day 6 post-fertilization) and their survival

depended on extracting resources from the host clutch. Finally, we further extended the duration of incubation to 14 days, which covers the period of host offspring release. At the end of incubation (day 4, 7 or 12 post-fertilization), mouthbrooded clutches were flushed from the female's mouth using a Pasteur pipette, and surviving *Synodontis* offspring were counted. For replicates with 14 days post-fertilization, all live offspring retained in the tank (protected under the false perforated bottom) were counted and scored as surviving.

The success or failure of development of *S. granulosus* in cichlid buccal cavities was tested using GLMM with binomial error distribution, with *Synodontis* species (*S. granulosus* as treatment, *S. multipunctatus* as control), duration of incubation and their interaction (to account for potential species-specific survival) as fixed factors. The interaction term was not significant ($P = 0.635$) and removed from the final model. Host female identity was used as random intercept to account for non-independence of groups of 2 or 4 *Synodontis* eggs incubated in the same host. The analyses of experiments with *S. granulosus* and *Shuja horei* and with *S. polli* and *Haplochromis* did not converge, likely due to zero survival in the treatment groups (0 of 18 *S. granulosus* eggs compared to 6 or 20 control cuckoo catfish eggs; and 0 of 20 *S. polli* eggs compared to 26 of 28 control cuckoo catfish eggs, respectively). While Fisher's exact test demonstrated significantly lower survival of *S. granulosus* in *S. horei* ($P = 0.021$) and of *S. polli* in *Haplochromis* ($P < 0.0001$), we do not report these analyses in the main text as the input data were more structured (grouped by host female individuals) than Fisher's exact test can handle. Instead, we declare zero survival of treatment embryos in mouthbrooding hosts which strongly contrasts with common survival of their respective controls.

All statistical tests were two-tailed. All individual measurements were taken from distinct individuals. Any covariates dropped from the final models are reported in the methods.

Data availability

All trophic, reproductive, ontogenetic and experimental data generated in this study have been deposited in the Figshare database [<https://doi.org/10.6084/m9.figshare.28646141>] and are freely available. Source data for figures are provided as a Source Data file. Raw ddRADseq reads have been deposited in NCBI's Sequence Read Archive (<https://www.ncbi.nlm.nih.gov/sra>) (BioProject: PRJNA1132910) under accession numbers SAMN42360897–SAMN42360967.

Code availability

The scripts to replicate all analyses reported in this study are available from Figshare [<https://doi.org/10.6084/m9.figshare.28646141>].

odstranil: results

odstranil: ¶

odstranil: Data associated with this manuscript are available in the supplementary materials and at Figshare (doi: 10.6084/m9.figshare.28646141).

odstranil: S

odstranil: (

odstranil: doi: 10.6084/m9.figshare.28646141)

References

1. Davies, N. *Cuckoos, Cowbirds and Other Cheats*. (AC Black, 2010).
2. Feeney, W. E. *et al.* Brood parasitism and the evolution of cooperative breeding in birds. *Science* **342**, 1506–1508 (2013).
3. Feeney, W. E., Welbergen, J. A. & Langmore, N. E. Advances in the study of coevolution between avian brood parasites and their hosts. *Annu. Rev. Ecol. Evol. Syst.* **45**, 227–246 (2014).
4. Kilner, R. M. & Langmore, N. E. Cuckoos versus hosts in insects and birds: adaptations, counter-adaptations and outcomes. *Biol. Rev.* **86**, 836–852 (2011).
5. Medina, I. & Langmore, N. E. The evolution of acceptance and tolerance in hosts of avian brood parasites: Evolution of defences in hosts. *Biol. Rev.* **91**, 569–577 (2016).
6. Thorogood, R., Spottiswoode, C. N., Portugal, S. J. & Gloag, R. The coevolutionary biology of brood parasitism: a call for integration. *Philos. Trans. R. Soc. B Biol. Sci.* **374**, 20180190 (2019).
7. Lyon, B. E. & Eadie, J. McA. Conspecific brood parasitism in birds: a life-story perspective. *Annu. Rev. Ecol. Evol. Syst.* **39**, 343–363 (2008).
8. Spottiswoode, C. N., Kilner, R. M. & Davies, N. B. Brood parasitism. In *The evolution of parental care* 226–356. (Oxford University Press, Oxford, 2012).
9. Sless, T. J. L., Danforth, B. N. & Searle, J. B. Evolutionary origins and patterns of diversification in animal brood parasitism. *Am. Nat.* **202**, 107–121 (2023).
10. Skelton, P. *Freshwater Fishes of Southern Africa*. (2024).
11. Sato, T. A brood parasitic catfish of mouthbrooding cichlid fishes in Lake Tanganyika. *Nature* **323**, 58–59 (1982).
12. Ronco, F. *et al.* Drivers and dynamics of a massive adaptive radiation in cichlid fishes. *Nature* **589**, 76–81 (2021).
13. Blažek, R. *et al.* Success of cuckoo catfish brood parasitism reflects coevolutionary history and individual experience of their cichlid hosts. *Sci. Adv.* **4**, eaar4380 (2018).
14. Wisenden, B. D. Alloparental care in fishes. *Rev. Fish Biol. Fisher.* **9**, 45–70 (1999).
15. Reichard, M. Cuckoo catfish. *Curr. Biol.* **29**, R722–R723 (2019).
16. Zimmermann, H., Blažek, R., Polačik, M. & Reichard, M. Individual experience as a key to success for the cuckoo catfish brood parasitism. *Nat. Commun.* **13**, 1723 (2022).
17. Cohen, M. S., Hawkins, M. B., Stock, D. W. & Cruz, A. Early life-history features associated with brood parasitism in the cuckoo catfish, *Synodontis multipunctatus* (Siluriformes: Mochokidae). *Philos. Trans. R. Soc. B Biol. Sci.* **374**, 20180205 (2019).
18. Zimmerman, H., Tolman, D. & Reichard, M. Low incidence of cannibalism among brood parasitic cuckoo catfish embryos. *Behav. Ecol.* **34**, 521–527 (2023).
19. Mouginot, P., Galipaud, M. & Reichard, M. The evolution of brood parasitism from host egg predation. *Behav. Ecol.* **35**, arae043 (2024).
20. Day, J. J. & Wilkinson, M. On the origin of the *Synodontis* catfish species flock from Lake Tanganyika. *Biol. Lett.* **2**, 548–552 (2006).
21. Koblmüller, S., Egger, B., Sturmbauer, C. & Sefc, K. M. Rapid radiation, ancient incomplete lineage sorting and ancient hybridization in the endemic Lake Tanganyika cichlid tribe Tropheini. *Mol. Phylogenet. Evol.* **55**, 318–334 (2010).

22. Englmaier, G. K. *et al.* Revised taxonomy of *Synodontis* catfishes (Siluriformes: Mochokidae) from the Lake Tanganyika basin reveals lower species diversity than expected. *Zool. J. Linn. Soc.* **202**, zlae130 (2024).
23. Koblmüller, S., Sturmbauer, C., Verheyen, E., Meyer, A. & Salzburger, W. Mitochondrial phylogeny and phylogeography of East African squeaker catfishes (Siluriformes: Synodontis). *BMC Evol. Biol.* **6**, 49 (2006).
24. Wright, J. J. & Page, L. M. Taxonomic revision of Lake Tanganyikan *Synodontis* (Siluriformes: Mochokidae). *Bull. Fla. Mus. Nat. Hist.* **46**, 57 (2006).
25. Peart, C. R., Bills, R., Newton, J., Near, T. J. & Day, J. J. Do sympatric catfish radiations in Lake Tanganyika show eco-morphological diversification? *Evol. J. Linn. Soc.* **3**, kzae015 (2024).
26. Schluter, D. *The Ecology of Adaptive Radiation*. (Oxford University Press, Oxford, 2000).
27. Stroud, J. T. & Losos, J. B. Ecological opportunity and adaptive radiation. *Annu. Rev. Ecol. Evol. Syst.* **47**, 507–532 (2016).
28. Reichard, M. *et al.* Lack of host specialization despite selective host use in brood parasitic cuckoo catfish. *Mol. Ecol.* **32**, 6070–6082 (2023).
29. Viertler, A., Salzburger, W. & Ronco, F. Comparative scale morphology in the adaptive radiation of cichlid fishes (Perciformes: Cichlidae) from Lake Tanganyika. *Biol. J. Linn. Soc.* **134**, (2021).
30. Jawad, L. A. Comparative morphology of scales of four teleost fishes from Sudan and Yemen. *J. Nat. Hist.* **39**, 2643–2660 (2005).
31. Muschick, M., Indermaur, A. & Salzburger, W. Convergent evolution within an adaptive radiation of cichlid fishes. *Curr. Biol.* **22**, 2362–2368 (2012).
32. Nair, P., Miller, C. M. & Fuiman, L. A. Tracing exploitation of egg boons: an experimental study using fatty acids and stable isotopes. *J. Exp. Biol.* **226**, jeb246247 (2023).
33. Hunter, J. R. & Kimbrell, C. A. Egg cannibalism in the Northern anchovy, *Engraulis mordax*. *Fish. Bull.* **78**, 811–816 (1980).
34. Yongo, E., Iteba, J. & Agember, S. Review of food and feeding habits of some *Synodontis* fishes in African freshwaters. *Oceanogr. Fish. Open Access J.* **10**, (2019).
35. Konings, A. *Tanganyika Cichlids in Their Natural Habitat*. (Cichlid Press, El Paso, 2015).
36. Stoddard, M. C. & Hauber, M. E. Colour, vision and coevolution in avian brood parasitism. *Philos. Trans. R. Soc. B Biol. Sci.* **372**, 20160339 (2017).
37. Santos, M. E. *et al.* The evolution of cichlid fish egg-spots is linked with a cis-regulatory change. *Nat. Commun.* **5**, 5149 (2014).
38. Cohen, M. S., Hawkins, M. B., Stock, D. W. & Cruz, A. Early life-history features associated with brood parasitism in the cuckoo catfish, *Synodontis multipunctatus* (Siluriformes: Mochokidae). *Philos. Trans. R. Soc. B Biol. Sci.* **374**, 20180205 (2019).
39. Atukorala, A. D. S. & Franz-Odenaal, T. A. Spatial and temporal events in tooth development of *Astyanax mexicanus*. *Mech. Dev.* **134**, 42–54 (2014).
40. Debiais-Thibaud, M. *et al.* Development of oral and pharyngeal teeth in the medaka (*Oryzias latipes*): comparison of morphology and expression of *evel* gene. *J. Exp. Zool. B Mol. Dev. Evol.* **308B**, 693–708 (2007).

41. Cohen, M. S., Hawkins, M. B., Knox-Hayes, J., Vinton, A. C. & Cruz, A. A laboratory study of host use by the cuckoo catfish *Synodontis multipunctatus*. *Environ. Biol. Fishes* **101**, 1417–1425 (2018).
42. Title, P. O. *et al.* The macroevolutionary singularity of snakes. *Science* **383**, 918–923 (2024).
43. Polačik, M., Reichard, M., Smith, C. & Blažek, R. Parasitic cuckoo catfish exploit parental responses to stray offspring. *Philos. Trans. R. Soc. B Biol. Sci.* **374**, 20180412 (2019).
44. Kattan, G. H. Shiny cowbirds follow the ‘shotgun’ strategy of brood parasitism. *Anim. Behav.* **53**, 647–654 (1997).
45. Payne, R. B. The evolution of clutch size and reproductive rates in parasitic cuckoos. *Evolution* **28**, 169–181 (1974).
46. Langmore, N. E. *et al.* Coevolution with hosts underpins speciation in brood-parasitic cuckoos. *Science* **384**, 1030–1036 (2024).
47. McClelland, S. C. *et al.* Highly virulent avian brood-parasitic species show elevated embryonic metabolic rates at specific incubation stages compared to less virulent and non-parasitic species. *Biol. Lett.* **20**, 20240411 (2024).
48. Spottiswoode, C. N. & Koorevaar, J. A stab in the dark: chick killing by brood parasitic honeyguides. *Biol. Lett.* **8**, 241–244 (2012).
49. Reichard, M. Zimmermann, H. Dataset and analytical code for ‘The ecological and developmental foundations of brood parasitism in a catfish’, Figshare. <https://doi.org/10.6084/m9.figshare.28646141> (2026).
50. Oksanen, J. vegan: Community Ecology Package. R package version 2.5-6. (2019).
51. Wang, Y., Naumann, U., Wright, S. T. & Warton, D. I. mvabund– an R package for model-based analysis of multivariate abundance data. *Methods Ecol. Evol.* **3**, 471–474 (2012).
52. Hogsden, K. L. & McHugh, P. A. Preservatives and sample preparation in stable isotope analysis of New Zealand freshwater invertebrates. *N. Z. J. Mar. Freshw. Res.* **51**, 455–464 (2017).
53. Kiljunen, M. *et al.* A revised model for lipid-normalizing $\delta^{13}\text{C}$ values from aquatic organisms, with implications for isotope mixing models. *J. Appl. Ecol.* **43**, 1213–1222 (2006).
54. McCutchan Jr, J. H., Lewis Jr, W. M., Kendall, C. & McGrath, C. C. Variation in trophic shift for stable isotope ratios of carbon, nitrogen, and sulfur. *Oikos* **102**, 378–390 (2003).
55. Post, D. M. Using Stable Isotopes to estimate trophic position: models, methods, and assumptions. *Ecology* **83**, 703–718 (2002).
56. Martinez Arabizu, P. pairwiseAdonis: Pairwise multilevel comparison using adonis. R package version 0.4, 1. (2020).
57. Dyková, I. *et al.* Oogenesis, spermatogenesis and spermiation structures in Lake Tanganyika *Synodontis* species (Mochokidae, Telostei: Siluriformes). *J. Vertebr. Biol.* **73**, (2024).
58. Brown-Peterson, N. J., Wyanski, D. M., Saborido-Rey, F., Macewicz, B. J. & Lowerre-Barbieri, S. K. A standardized terminology for describing reproductive development in fishes. *Mar. Coast. Fish.* **3**, 52–70 (2011).

odstranil: Dataset

odstranil: 2025

nastavil formátování: Němčina (Rakousko)

59. Sipos, M. J. *et al.* Evaluation of cGnRH IIa for induction spawning of two ornamental *Synodontis* species. *Aquaculture* **511**, 734226 (2019).
60. Troscianko, J. & Stevens, M. Image calibration and analysis toolbox – a free software suite for objectively measuring reflectance, colour and pattern. *Methods Ecol. Evol.* **6**, 1320–1331 (2015).
61. Brooks, M. E. {glmmTMB} Balances Speed and Flexibility Among Packages for Zero-inflated Generalized Linear Mixed Modeling. *R J.* **9**(2):378–400. (2017).
62. Hartig, F. DHARMA: Residual diagnostics for hierarchical (multi-level/mixed) regression models. R Package Version 0.4. 4. (2021).
63. Lenth R V., Piaskowski J. *emmeans: Estimated Marginal Means, aka Least-Squares Means. R package version 2.0.1 (2025)*.
64. Revell, L. Phytools 2.0: an updated R ecosystem for phylogenetic comparative methods (and other things). *PeerJ*, **12**, e16505 (2024).
65. Harmon, L. J., Weir, J. T., Brock, C. D., Glor, R. E. & Challenger, W. GEIGER: investigating evolutionary radiations. *Bioinformatics* **24**, 129–131 (2008).
66. Walker, M. & Kimmel, C. A two-color acid-free cartilage and bone stain for zebrafish larvae. *Biotech. Histochem.* **82**, 23–28 (2007).

nastavil formátování: Písmo: (výchozí) Times New Roman

Naformátování: Normální, Odsazení: Vlevo: 0 cm, Předšazení: 0,68 cm, Řádkování: jednoduché

nastavil formátování: Písmo:

odstranil: 63

odstranil: 64

odstranil: 65

Acknowledgments We thank the Department of Fisheries in Mpulungu, especially Lwabanya Mabo, for facilitating research in Zambia, and Tanzanian Fisheries Research Institute in Kigoma, especially Deogratias P. Mulokozi, for facilitating research in Tanzania. The manuscript benefited from comments from Carl Smith, Lukáš Kratochvíl, Miranda Sherlock and four anonymous referees. This research was funded by the Czech Science Foundation (21-00788X to M.R.) and, in part, by the Austrian Science Fund (FWF) (doi.org/10.55776/J4584 to H.Z). For the purpose of open access, the authors have applied a CC BY public copyright license to any Author Accepted Manuscript version arising from this submission.

odstranil: In Zambia, research was conducted under research permit K-4335/18 KA/K.48/18 and individual study permits of MR, HZ, RB, MP, VB and SK. In Tanzania, samples were collected under research permits from COSTECH (2022-204-NA-2022-228) and TAFIRI (TAFIRI/HQ/RES.CLEARANCE/82).

odstranil: Research adhered to all national and institutional animal care and use guidelines (permit No. CZ62760203).

Author Contribution Statement:

Conceptualization: MR, SK, HZ

Methodology: MR, HZ, RB, KP, JŽ, RC

Investigation: MR, HZ, RB, TS, MP, GE, VB, KP, JŽ, LK, ID, SK

Visualization: MR, HZ, TS, GE

Funding acquisition: MR, HZ

Project administration: MR

Supervision: MR

Writing – original draft: MR, HZ

Writing – review and editing: MR, HZ, RB, TS, MP, GE, VB, KP, JŽ, LK, ID, RC, SK

Competing Interests Statement

The authors declare that they have no competing interests.

Figure Captions

Fig. 1 | Phylogenetic relationships, dietary, and trophic separation among LT *Synodontis*.

a, The maximum likelihood phylogenetic tree of the studied LT *Synodontis* flock, based on the concatenated alignment of 6536 ddRADseq loci and 5000 ultrafast bootstrap replicates ²⁶. **b**, The diet composition of LT *Synodontis* expressed as the volumetric proportion of specific dietary categories in the gastrointestinal tract (n=122; species-specific sample size is provided above each bar). **c**, A biplot from the principal coordinates analysis (PCoA from the 'vegan' R package) comparing diet composition across LT *Synodontis*, derived from Bray-Curtis dissimilarity matrices. Ellipses represent the 95% confidence intervals for each species. Black dots denote the positions of key diet components that differentiate the species-specific diet. **d**, Iso-space plots of LT *Synodontis* (n=198) plotted over mean (95% CI) values of baseline items (n=34). Values are reported in standard per-mil notation (‰); long-term analytical precision was 0.2‰ for $\delta^{13}\text{C}$ values and 0.1‰ for $\delta^{15}\text{N}$ values. **e**, Overlap between coexisting LT *Synodontis* (95% CI ellipses) and selected cichlid species (mean and 95% CI error bars) (n=62). Source data are provided as a Source Data file.

Fig. 2 | Reproductive parameters of LT and riverine *Synodontis*. **a**, Clutch size, measured as the number of ripe eggs extracted from female gonads following artificial reproduction via hormonal stimulation (n = 80, two-sided LM, $\chi^2 = 297$, df = 7, P < 0.001). **b**, Egg size (measured along the longest axis of IVF-produced freshly spawned eggs) (n = 340 eggs from 34 clutches, two-sided LMM, $\chi^2 = 2010$, df = 7, P < 0.001). **c**, Ovary mass (grams, log-transformed) (n = 127, GLM, $\chi^2 = 21.3$, df = 3, P < 0.001) and **d**, Testes mass, expressed as a product of fish body size (mm) in individuals with mature gonads (n = 118, GLM, $\chi^2 = 4.8$, df = 3, P = 0.185). Raw data points, best fit linear models with 95% confidence intervals are displayed. **e**, Egg coloration, measured using cone-catch images transformed into CIELab color space, with the first two axes representing green-red (CIE-A) and blue-yellow (CIE-B). Insets on the left side display photographs of the eggs of *S. granulosus* and *S. multipunctatus*. **f**, Egg adhesion, expressed as a percentage of freshly laid eggs remaining adhered to a vertical glass surface (n = 35 clutches, ordered-beta GLM, $\chi^2 = 119$, df = 7, P < 0.001). The box plots show median (line), 25th and 75th percentiles (box), and 1.5 × the interquartile range (whiskers). Different letters in a-c, and f indicate statistically different values (Post-hoc tests with Benjamini-Hochberg correction, P < 0.05; Tables S7-S10). No further adjustments for multiple comparisons were made. All replicates were biological, not technical, replicates. Source data are provided as a Source Data file.

Fig. 3 | Early development of *Synodontis* embryos. **a**, Growth (notochord length) of seven *Synodontis* species (visualized as a linear model fitted for post-hatching growth with 95% confidence intervals and raw data points, 2-10 days post-fertilization, n = 558). Source data are provided as a Source Data file. **b**, The appearance of all LT (and one outgroup) *Synodontis* species at the age of 10 days, presented to the same scale. First appearance (arrow) and mineralization (red color) of **c**, oral and **d**, pharyngeal dentition in seven *Synodontis* species (scale bar in each inset photo). The complete series of pharyngeal dentition ontogeny is shown in Figure S4.

odstranil: Asterisks indicate whether clutches of the cuckoo catfish were significantly smaller than those of all other species (LM, P < 0.001).

nastavil formátování: Písmo: Symbol

nastavil formátování: horní index

odstranil: Statistically significant differences in pairwise contrasts with *S. multipunctatus* are indicated by asterisks.

odstranil: B

odstranil: Values significantly different from those estimated for the cuckoo catfish are indicated by asterisks.

odstranil: Sidak p

odstranil: Supplementary

odstranil: XXX - XXX

odstranil: Extended Data

odstranil: 2

odstranil: .

Fig. 4 | Potential for brood parasitism in *S. granulosus* and *S. polli*. Proportion of surviving embryos of *S. granulosus*, *S. polli* and their respective cuckoo catfish controls in an evolutionary naive non-sympatric (non-rejecting) host cichlid (*Haplochromis* CH44 from Lake Victoria with an egg diameter of 2.8 mm) over **a**, the period of 4-7 days (**n = 20 eggs across 6 clutches for *S. granulosus* and 28 eggs across 7 clutches in the cuckoo catfish controls**), **b**, 12-14 days (**n = 36 eggs across 9 clutches for *S. granulosus* and 24 eggs across 6 clutches in the cuckoo catfish controls**) and **c**, the period of 7 days for *S. polli* treatment (**n = 40 eggs across 10 clutches for *S. polli* and 56 eggs across 14 clutches in the cuckoo catfish controls**). **d**, A cuckoo catfish embryo consuming host offspring, with the host's eye visible in the catfish's stomach. Seven-day-old offspring recovered from host mouth of **e**, cuckoo catfish with fully filled guts and **f**, *S. granulosus* with sparse remnants of yolk in their alimentary tracts. **g**, Proportion of surviving embryos in a Lake Tanganyika (able to respond to cuckoo catfish parasitism) cichlid host (*Shuja horei*, egg diameter 4.7 mm) over the period of 4-7 days (**n = 18 eggs across 6 clutches for *S. granulosus* and 20 eggs across 6 clutches in the cuckoo catfish controls**). Bars and whiskers represent overall estimated means and one standard errors, while the individual points illustrate the proportions of surviving *Synodontis* embryos in particular cichlid clutches. Source data are provided as a Source Data file.

nastavil formátování: Písmo: Kurzíva

odstranil: and

nastavil formátování: Písmo: Kurzíva

nastavil formátování: Písmo: Kurzíva

odstranil: c

odstranil: d

nastavil formátování: Písmo: Tučné

odstranil: e

odstranil: f